

# Hyperfine spectroscopy in a quantum-limited spectrometer

Sebastian Probst[1], Gengli Zhang[2], Miloš Rančić[1], Vishal Ranjan[1], Marianne Le Dantec[1], Zhonghan Zhang[3], Bartolo Albanese[1], Andrin Doll[4], Ren Bao Liu[2], John Morton[5], Thierry Chanelière[6], Philippe Goldner[3], Denis Vion[1], Daniel Esteve[1], and Patrice Bertet[1]

[1]Quantronics group, SPEC, CEA, CNRS, Université Paris-Saclay, CEA Saclay 91191 Gif-sur-Yvette Cedex, France
[2]Department of Physics and The Hong Kong Institute of Quantum Information Science and Technology, The Chinese University of Hong Kong, Shatin, New Territories, Hong Kong, China
[3]Chimie ParisTech, PSL University, CNRS, Institut de Recherche de Chimie Paris, 75005 Paris, France
[4]Laboratory of nanomagnetism and oxides, SPEC, CEA, CNRS, Université Paris-Saclay, CEA Saclay 91191 Gif-sur-Yvette Cedex, France
[5]London Centre for Nanotechnology, University College London, London WC1H 0AH, United Kingdom
[6]Univ. Grenoble Alpes, CNRS, Grenoble INP, Institut Néel, 38000 Grenoble, France

**Abstract.** We report measurements of electron spin echo envelope modulation (ESEEM) performed at millikelvin temperatures in a custom-built high-sensitivity spectrometer based on superconducting micro-resonators. The high quality factor and small mode volume (down to 0.2pL) of the resonator allow to probe a small number of spins, down to $5 \cdot 10^2$. We measure 2-pulse ESEEM on two systems: erbium ions coupled to $^{183}$W nuclei in a natural-abundance $CaWO_4$ crystal, and bismuth donors coupled to residual $^{29}$Si nuclei in a silicon substrate that was isotopically enriched in the $^{28}$Si isotope. We also measure 3- and 5-pulse ESEEM for the bismuth donors in silicon. Quantitative agreement is obtained for both the hyperfine coupling strength of proximal nuclei, and the nuclear spin concentration.

07.57.Pt,76.30.-v,85.25.-j

## 1 Introduction

Electron paramagnetic resonance (EPR) spectroscopy provides a set of versatile tools to study the magnetic environment of unpaired electron spins (Schweiger and Jeschke, 2001). Most EPR spectrometers rely on the inductive detection of the spin signal by a microwave resonator tuned to the spin Larmor frequency. They achieve a rather limited spin sensitivity (between $10^6$ and $10^{11}$ spin/$\sqrt{\text{Hz}}$, depending on the frequency used and the temperature). Efforts to enhance the spin sensitivity have turned to alternative detection methods, requiring dedicated instruments or specific samples. Electrical (Elzerman et al., 2004; Veldhorst et al., 2014; Morello et al., 2010; Pla et al., 2012) and optical (Jelezko et al., 2004) detection of spin resonance as well as scanning-probe methods have reached sufficient sensitivity to detect individual electron spins.

In parallel, recent results have shown that the inductive detection method can also be pushed to much higher sensitivity than previously achieved, using concepts and techniques borrowed from research on superconducting quantum circuits. An inductive-detection spectrometer relying on a superconducting planar micro-resonator combined with a Josephson Parametric Amplifier (JPA), cooled down to millikelvin temperatures, has achieved a sensitivity of 65 spin/$\sqrt{\text{Hz}}$ for detecting Hahn echoes originating from donors in silicon (Probst et al., 2017). A particular feature of the spectrometer is that the output noise is governed by quantum fluctuations of the microwave field at low temperatures, with negligible contribution of thermal fluctuations.

Hahn echoes are, however, the simplest pulse sequence used in EPR spectroscopy. They are useful to determine the electron spin density as well as the spin Hamiltonian parameters and their distribution. But the richness of EPR spectroscopy also





comes from the ability to characterize the local magnetic environment of the electron spins, often consisting of a set of nuclear spins or of other electron spins. For that, hyperfine spectroscopy is required, which uses more elaborate pulse sequences and requires larger detection bandwidth. Previous hyperfine spectroscopy measurements with superconducting micro-resonators include the electron-nuclear double resonance detection of donors in silicon (Sigillito et al., 2017) and the electron-spin-echo
envelope modulation (ESEEM) of erbium ions by the nuclear spin of yttrium in a $Y_2SiO_5$ crystal (Probst et al., 2015).

Here, we demonstrate that hyperfine spectroscopy is compatible with quantum-limited EPR spectroscopy despite its additional requirements in terms of pulse complexity and bandwidth, by measuring ESEEM in two model electron spin systems. We measure the ESEEM of erbium ions coupled to $^{183}W$ nuclei in a scheelite crystal ($CaWO_4$) with a simple two-pulse sequence, and get quantitative agreement with a simple dipolar interaction model. We also measure the ESEEM of bismuth donors in
silicon caused by $^{29}Si$ nuclei using 2, 3, and 5-pulse sequences (Schweiger and Jeschke, 2001; Kasumaj and Stoll, 2008). Compared to other ESEEM measurements on donors in silicon (Witzel et al., 2007; Abe et al., 2010), ours are performed in an isotopically purified sample having a 100 times lower concentration in $^{29}Si$ (500 ppm) than natural abundance. As a result, the dominant hyperfine interactions in the ESEEM signal are very low (on the order of $100\,\mathrm{Hz}$) and have to be detected at low magnetic fields (around $0.1\,\mathrm{mT}$). These results bring quantum-limited EPR spectroscopy one step closer to real-world
applications.

## 2   ESEEM spectroscopy : theory

### 2.1   Phenomenology

We start by briefly discussing the ESEEM phenomenon. Consider an ensemble of electron spins placed in a magnetic field $B_0$. The spin ensemble linewidth $\Gamma$ is broadened by a variety of mechanisms : spatial inhomogeneity of the applied field $B_0$, local
magnetic fields generated by magnetic impurities throughout the sample, and spatially inhomogeneous strain or electric fields. One prominent way to mitigate the effect of this inhomogeneous broadening is the spin-echo sequence (also called Hahn echo, or two-pulse echo). It consists of a $\pi/2$ pulse at time $t = 0$ and a $\pi$ pulse after a delay $\tau$ (see Fig.1a). This $\pi$ pulse reverses the evolution of the phase of the precessing magnetic dipoles, which leads at a later time $2\tau$ to their refocussing and the emission of a microwave pulse (the echo) of amplitude $V_{2\mathrm{p}}(\tau)$.

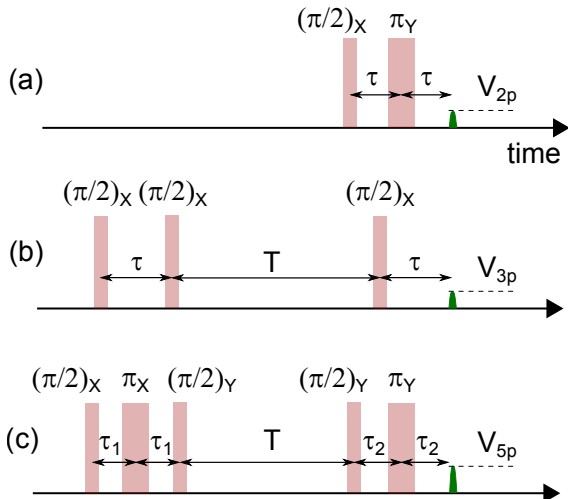

**Figure 1.** Sequences used for 2-pulse (a), 3-pulse (b), and 5-pulse (c) ESEEM measurements.

In general, $V_{2\mathrm{p}}(\tau)$ decays monotonically; it can however also display oscillations. Such ESEEM was first observed by Mims and co-workers (Mims et al., 1961; Rowan et al., 1965) for $Ce^{3+}$ ions in a $CaWO_4$ crystal, and was interpreted as being caused by the dipolar interaction of the electronic spin of the $Ce^{3+}$ ions with the $^{183}W$ nuclear spins of the crystal. The oscillation frequencies appearing in the ESEEM pattern are related to the nuclear spin Larmor frequencies and to their coupling to the



electron spin. As such, ESEEM measurements provide spectroscopic information on the nature of the nuclear spin bath and its density, and ESEEM spectroscopy has become an essential tool in advanced EPR (Schweiger and Jeschke, 2001; Mims et al., 1990). ESEEM has also been observed for individual spins measured optically, in particular for individual NV centers in diamond coupled to a bath of $^{13}$C nuclear spins (Childress et al., 2006). A more complete theory of ESEEM is presented in (Mims, 1972). Our goal here is to provide a simple picture of the physics involved, as well as to introduce useful formulas and notations.

### 2.2 Two-spin-1/2 model

We follow the analysis in Ref.(Schweiger and Jeschke, 2001) of the model case depicted in Fig.2a. An electron spin $S = 1/2$, with an isotropic g-tensor, is coupled to a proximal nuclear spin $I = 1/2$. Both are subject to a magnetic field $B_0$ applied along $z$. The system Hamiltonian is

$$H_0 = H_e + H_n + H_{hf}, \tag{1}$$

where $H_e = \omega_S S_z$ ($H_n = \omega_I I_z$) is the Zeeman Hamiltonian of the electron (nuclear) spin with Larmor frequency $\omega_S$ ($\omega_I$), and $H_{hf}$ is the electron-nuclear hyperfine interaction, which includes their dipole-dipole coupling and may include a Fermi contact term as well. We assume that $\omega_S$ is much larger than the hyperfine interaction strength, in which case terms proportional to the $S_x$ and $S_y$ operators can be neglected. This secular approximation leads to a hyperfine Hamiltonian of the form $H_{hf} = AS_z I_z + BS_z I_x$, with the expressions for $A$ and $B$ depending on the details of the hyperfine interaction(Schweiger and Jeschke, 2001).

Overall, the system Hamiltonian is

$$H_0 = \omega_S S_z + \omega_I I_z + AS_z I_z + BS_z I_x. \tag{2}$$

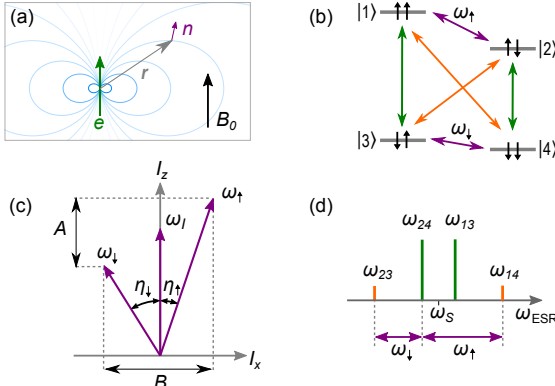

**Figure 2.** ESEEM model system for electron spin $S = 1/2$ and nuclear spin $I = 1/2$ with $\omega_I, A, B > 0$. (a) Nuclear spin (purple) subject to external field $B_0$ and dipole field (blue) of a nearby electron spin (green) located at relative position $\boldsymbol{r}$. (b) Energy diagram showing the electron transitions (green), the nuclear transitions (purple), and the (normally forbidden) electro-nuclear transitions (orange). The energy levels $|1\rangle$, ..., $|4\rangle$ are labeled according to the eigenstates of the Zeeman basis. (c) Quantization axes $\omega_\uparrow$ and $\omega_\downarrow$ due to mixing of the nuclear states, which results in inclination of the quantization axis from $z$ by the angles $\eta_\uparrow$ and $\eta_\downarrow$, respectively. (d) EPR spectrum showing the electron transitions (green) and the electro-nuclear transitions (orange) as well as the relation of these ESR transitions to the nuclear frequencies $\omega_\uparrow$ and $\omega_\downarrow$ (purple).

Because of the $BS_z I_x$ term, the nuclear spin is subjected to an effective magnetic field whose direction (and magnitude) depend on the electron spin state $|\uparrow_e\rangle$ or $|\downarrow_e\rangle$. Its eigenstates therefore depend on the electron spin state, so that nuclear-spin-non-preserving transitions become allowed, which leads to the ESEEM phenomenon.

More precisely, the Hamiltonian Eq.2 can be diagonalized leading to the following four eigenstates




**4** : 

$$
\begin{aligned}
|1\rangle &= |\uparrow_e\rangle(\cos\frac{\eta_\uparrow}{2}|\uparrow_n\rangle + \sin\frac{\eta_\uparrow}{2}|\downarrow_n\rangle) \\
|2\rangle &= |\uparrow_e\rangle(\sin\frac{\eta_\uparrow}{2}|\uparrow_n\rangle - \cos\frac{\eta_\uparrow}{2}|\downarrow_n\rangle) \\
|3\rangle &= |\downarrow_e\rangle(\cos\frac{\eta_\downarrow}{2}|\uparrow_n\rangle + \sin\frac{\eta_\downarrow}{2}|\downarrow_n\rangle) \\
|4\rangle &= |\downarrow_e\rangle(\sin\frac{\eta_\downarrow}{2}|\uparrow_n\rangle - \cos\frac{\eta_\downarrow}{2}|\downarrow_n\rangle),
\end{aligned}
\tag{3}
$$

5 where subscript e (resp. n) refers to the electron (resp. nuclear) state, and

$$
\begin{aligned}
\eta_\uparrow &= \arctan\frac{B}{A + 2\omega_I} \\
\eta_\downarrow &= \arctan\frac{B}{A - 2\omega_I}.
\end{aligned}
\tag{4}
$$

Physically, $\eta_{\uparrow,\downarrow}$ is the electron-spin-state-dependent angle between the effective magnetic field seen by the nuclear spin and the quantization axis $z$. The energies of these states are

$$
\begin{aligned}
10\ \epsilon_1 &= \frac{\omega_S}{2} + \frac{\omega_\uparrow}{2} \\
\epsilon_2 &= \frac{\omega_S}{2} - \frac{\omega_\uparrow}{2} \\
\epsilon_3 &= -\frac{\omega_S}{2} + \frac{\omega_\downarrow}{2} \\
\epsilon_4 &= -\frac{\omega_S}{2} - \frac{\omega_\downarrow}{2},
\end{aligned}
\tag{5}
$$

with

$$
\begin{aligned}
15\ \omega_\uparrow &= (\omega_I + \frac{A}{2})\cos\eta_\uparrow - \frac{B}{2}\sin\eta_\uparrow \\
\omega_\downarrow &= (\omega_I - \frac{A}{2})\cos\eta_\downarrow - \frac{B}{2}\sin\eta_\downarrow.
\end{aligned}
$$

Levels $|1\rangle$ and $|2\rangle$ (resp. $|3\rangle$ and $|4\rangle$) belong to the $|\uparrow_e\rangle$ (resp. $|\downarrow_e\rangle$) subspace, so that $\omega_{12}$ (resp. $\omega_{34}$) can be seen as the nuclear frequency when the electron is in $|\uparrow_e\rangle$ (resp. in $|\downarrow_e\rangle$).

The transition amplitude between pairs of levels is proportional to the matrix element of operator $S_x$. It is easily shown that
$\langle 1|S_x|3\rangle = \langle 2|S_x|4\rangle = \cos\eta$, while $\langle 1|S_x|4\rangle = -\langle 2|S_x|3\rangle = \sin\eta$, with $\eta = (\eta_\uparrow - \eta_\downarrow)/2$. When the angles $\eta_\uparrow, \eta_\downarrow$ are equal or nearly equal, only the nuclear-spin preserving transitions $1 \leftrightarrow 3$ and $2 \leftrightarrow 4$ are allowed since $\sin\eta \simeq 0$; this occurs either when $B = 0$ (due to a specific orientation of the dipolar field, or to a purely isotropic hyperfine coupling), or when $B \neq 0$ but $\omega_I \gg A$ (very weak coupling limit) or $\omega_I \ll A$ (very strong coupling limit). On the contrary, when the direction of the effective magnetic field seen by the nuclear spin is electron-spin-dependent, all 4 transitions become allowed since $\sin\eta \simeq \cos\eta \simeq 1/\sqrt{2}$. This
occurs when $B \neq 0$ and $\omega_I \simeq \pm A/2$.

### 2.3 Multi-pulse ESEEM

Because of the level structure shown in Fig.2, and assuming for simplicity microwave pulses so short that their bandwidth is much larger than $\omega_{\uparrow,\downarrow}$, microwave pulses at the electron spin frequency $\omega_S$ excite the allowed transitions $1 \leftrightarrow 3$ and $2 \leftrightarrow 4$, but also the normally forbidden $1 \leftrightarrow 4$ and $2 \leftrightarrow 3$, leading to coherence transfer between the levels and to beatings. Note that for
simplicity we assume that the microwave pulses are ideal and so short that their bandwidth is much larger than $\omega_{12}$ and $\omega_{34}$.

It is then possible to compute analytically the effect of a two-pulse echo sequence consisting of an instantaneous ideal $\pi/2$ pulse and an instantaneous ideal $\pi$ pulse (see Fig.1), disregarding any decoherence. The resulting echo amplitude (Schweiger and Jeschke, 2001) is given by





$$V_{2\mathrm{p}}(\tau) = 1 - \frac{k}{4}[2 - 2\cos(\omega_\uparrow\tau) - 2\cos(\omega_\downarrow\tau)$$
$$+ \cos((\omega_\uparrow - \omega_\downarrow)\tau) + \cos((\omega_\uparrow + \omega_\downarrow)\tau)], \tag{6}$$

with

$$k = \left[\frac{B\omega_I}{\omega_\uparrow\omega_\downarrow}\right]^2. \tag{7}$$

The spin-echo amplitude is modulated by a function whose frequency spectrum and amplitude contain information about the nuclear spin Larmor frequency $\omega_I$ as well as its hyperfine coupling $(A, B)$ to the electron spin. The modulation contrast $0 \le k \le 1$ is maximal when transitions 1-4 and 2-3 are maximally allowed, corresponding to $\omega_I \simeq A/2$.

The above results are exact, as long as the secular approximation is valid and the pulses are ideal. In the weak-coupling limit $A, B \ll \omega_I, \omega_\uparrow \simeq \omega_\downarrow \simeq \omega_I$ so that $V_{2\mathrm{p}}(\tau) = 1 - \frac{k}{4}[3 - 4\cos(\omega_I\tau) + \cos(2\omega_I\tau)]$, with $k = (B/\omega_I)^2 \ll 1$. In this limit, the echo modulation spectrum directly yields the nuclear spin Larmor frequency, and also contains components at twice this frequency. Note however that in practice, the $\pi$ pulse bandwidth is always finite, because of the resonator bandwidth or limited pulse power; this sets a limit to the range of detectable modulation frequencies.

The electron spin is often coupled to $N$ nuclear spins, with $N > 1$. Since all nuclear spin subspaces can be diagonalized separately, the total ESEEM modulation is simply given by the product of each nuclear spin modulation $V_{2p,l}(\tau)$, $l$ being the nuclear spin index. Taking also into account that the electron spin is also subject to decoherence processes, modelled for instance by an exponential decay with time constant $T_2$, the echo envelope is

$$V'_{2\mathrm{p}}(\tau) = \exp\left(-2\tau/T_2\right) \prod_{l=1}^{N} V_{2p,l}(\tau). \tag{8}$$

The modulation pattern $V'_{2\mathrm{p}}(\tau)$ yields quantitative information about the nature and coupling of the nuclear spins surrounding the electron spin whose echo is measured, and is therefore a useful tool in EPR spectroscopy. When the environmental nuclei have a certain probability $p$ to be of a given isotope with a nuclear spin $I = 1/2$, and a probability $1 - p$ to be of an isotope with $I = 0$, the above formulas are straightforwardly modified (Rowan et al., 1965) by writing

$$V_{2\mathrm{p},l}(\tau) = 1 - \frac{pk_l}{4}[2 - 2\cos(\omega_{\uparrow,l}\tau) - 2\cos(\omega_{\downarrow,l}\tau)$$
$$+ \cos((\omega_{\uparrow,l} - \omega_{\downarrow,l})\tau) + \cos((\omega_{\uparrow,l} + \omega_{\downarrow,l})\tau)]. \tag{9}$$

The echo signal $V'_{2\mathrm{p}}(\tau)$ is the sum of terms that have the general form $p^L \prod_{l=1}^{l=L} k_l \cos(\omega_{\mu,l}\tau)$, where $l$ runs over a subset of $L$ nuclei and $\mu = \uparrow, \downarrow$. If $p \ll 1$, this expression is well approximated by keeping only the $L = 1$ terms, which then yields

$$V_{2\mathrm{p}}(\tau) \simeq 1 - \sum_{l=1}^{l=N} \frac{pk_l}{4}[2 - 2\cos(\omega_{\uparrow,l}\tau) - 2\cos(\omega_{\downarrow,l}\tau)$$
$$+ \cos((\omega_{\uparrow,l} - \omega_{\downarrow,l})\tau) + \cos((\omega_{\uparrow,l} + \omega_{\downarrow,l})\tau)]. \tag{10}$$

One limitation of the previous pulse sequence is that the modulation envelope can only be measured up to a time of order $T_2$ due to electron spin decoherence, which may be too short for appreciable spectral resolution. This limitation can be overcome by the three-pulse echo sequence shown in Fig. 1b. It consists of a $\pi/2$ pulse applied at $t = 0$ followed, after a time $\tau$ chosen such that $\tau < T_2$, by a second $\pi/2$ pulse. After a variable delay $T$, a third $\pi/2$ pulse is applied, leading to the emission of a stimulated echo at time $t = T + 2\tau$. The interest of this sequence is that the first pair of $\pi/2$ pulses generates nuclear spin coherence that can survive up to the nuclear spin coherence time $T_{2,\mathrm{n}}$ which is in general much longer than $T_2$ (and close to the electron energy spin relaxation time $T_1$). An analytical formula can be derived for the three-pulse echo amplitude in the ideal pulse approximation (Schweiger and Jeschke, 2001)



$$
\begin{aligned}
V_{3\mathrm{p}}(T) \quad = \quad & \exp(-T/T_{2,\mathrm{n}})\exp(-2\tau/T_2) \\
& \{1 - \frac{k}{4}[[1-\cos\omega_\downarrow\tau][1-\cos\omega_\uparrow(T+\tau)] \\
& + [1-\cos\omega_\uparrow\tau][1-\cos\omega_\downarrow(T+\tau)]]\}.
\end{aligned}
\tag{11}
$$

Contrary to two-pulse ESEEM, three-pulse echo modulation as a function of $T$ only contains the $\omega_\downarrow, \omega_\uparrow$ frequency compo-
nents, and not their sum or difference; that is, in the weak-coupling limit $A, B \ll \omega_I$, only the nuclear spin Larmor frequency
$\omega_I$ appears in the spectrum. Another difference is that the modulation pattern and amplitude depend on $\tau$; in particular, its
amplitude is zero whenever $\omega_{\downarrow,\uparrow}\tau = 2\pi n$ with $n$ integer (*blind spots*).

For weakly coupled nuclei, the modulation amplitude of 3-pulse ESEEM can be enhanced by up to one order of magnitude
by using a more complex pulse sequence known as 5-pulse ESEEM (Schweiger and Jeschke, 2001; Kasumaj and Stoll, 2008),
and shown in Fig.1. The analytical formula for the five-pulse echo amplitude $V_{5\mathrm{p}}$ is given in the Supplementary Information.

Equation 8, with proper modification to take into account contributions of different pathways, can be applied to the 3-
and 5-pulse ESEEM to treat coupling to multiple nuclear spins. The details are shown in Section III.C of the Supplementary
Information.

### 2.4 Fictitious spin model

The electronic spins that we consider in this work involve an unpaired electron with spin $S_0 = 1/2$ either located around or
trapped by an ionic defect, which itself can possess a non-zero nuclear spin $I_0$. These two spins of the defect are strongly
coupled and form therefore a multi-level system, which can nevertheless be mapped to an effective, fictitious, spin-1/2 model
as explained below (Schweiger and Jeschke, 2001), to which the model of Section IIC can be applied.

The system spin Hamiltonian writes

$$
H_{\mathrm{ion}} = \beta_e \boldsymbol{B_0} \cdot \bar{\mathbf{g}}_e \cdot \boldsymbol{S_0} + \boldsymbol{S_0} \cdot \bar{\mathbf{A}}_0 \cdot \boldsymbol{I_0},
\tag{12}
$$

Here, $\beta_e$ is the electron Bohr magneton, $\bar{\mathbf{g}}_e$ is the (possibly anisotropic) gyromagnetic tensor, and $\bar{\mathbf{A}}_0$ the hyperfine tensor. The
nuclear Zeeman interaction of the defect system, being small compared to the hyperfine interaction in the range of magnetic
fields explored here, is neglected from the Hamiltonian.

This multi-level electron-spin system is coupled to other nuclear spins in the lattice, giving rise to ESEEM. Consider a
nuclear spin at a lattice site $j$, defined by its location $\boldsymbol{r}_j$ with respect to the electron spin. The nuclear Zeeman Hamiltonian
is $H_j = \omega_I I_{j,z}$, with $\omega_I = g_n \beta_n B_0$, $g_n$ being the nuclear g-factor and $\beta_n$ the nuclear magneton. Its hyperfine coupling to the
electron spin system is described by the Hamiltonian

$$
H_{j,\mathrm{hf}} = \boldsymbol{S_0} \cdot \bar{\mathbf{A}}_j \cdot \boldsymbol{I_j},
\tag{13}
$$

with

$$
\bar{\mathbf{A}}_j = \bar{\mathbf{A}}_{j,\mathrm{cf}} + \bar{\mathbf{A}}_{j,\mathrm{dd}}.
\tag{14}
$$

This hyperfine tensor consists of a Fermi contact term $\bar{\mathbf{A}}_{j,\mathrm{cf}} = \frac{2}{3}\mu_0 \beta_e g_n \beta_n \bar{\mathbf{g}}_e |\psi(\mathbf{r}_j)|^2$ and a dipole-dipole term $\bar{\mathbf{A}}_{j,\mathrm{dd}} = \frac{3\mu_0}{4\pi|\boldsymbol{r}_j|^5} \beta_e \beta_n g_n [r_j^2 \boldsymbol{g}_e - 3(\boldsymbol{g}_e \cdot \boldsymbol{r}_j)\boldsymbol{r}_j]$, $\psi(\mathbf{r}_j)$ being the electron wavefunction at the nuclear spin location.

The Hamiltonian $H_{\mathrm{ion}}$ (Eq.12) can be diagonalized, yielding $4I_0 + 2$ energy levels. It is in general possible to isolate two
levels $|\alpha\rangle$ and $|\beta\rangle$ that are coupled by an ESR-allowed transition and are resonant or quasi-resonant with the microwave cavity,
with a transition frequency $\omega_S$. If these two levels are sufficiently separated in energy from other levels of $H_{\mathrm{ion}}$, they define a
fictitious $S = 1/2$ system. Writing the total Hamiltonian $H_{\mathrm{ion}} + H_j + H_{\mathrm{hf},j}$ restricted to this two-dimensional subspace yields

$$
\begin{aligned}
H_0 \quad = \quad & \omega_S S_z + (\omega_I + \frac{m_S^\alpha + m_S^\beta}{2} A_{j,zz}) I_{j,z} \\
& + \frac{m_S^\alpha + m_S^\beta}{2} A_{j,zx} I_{j,x} \\
& + (m_S^\alpha - m_S^\beta)(A_{j,zz} S_z I_{j,z} + A_{j,zx} S_z I_{j,x})
\end{aligned}
\tag{15}
$$





where $m_S^{\alpha,\beta} = \langle \alpha, \beta | S_{0,z} | \alpha, \beta \rangle$.

Equation 15 maps the more complex system to the simple model of section IIB. Compared to Eq. (2), two differences appear. First, the hyperfine interaction parameters $A, B$ are rescaled by the effective longitudinal magnetization difference $(m_S^\alpha - m_S^\beta)$ which depends on the two levels considered. Second, when the average longitudinal magnetization of the two levels $(m_S^\alpha + m_S^\beta)$ is non-zero, the nuclear spin sees an extra Zeeman contribution which may be tilted with respect to the $z$ axis. Once taken into account these corrections, the analysis and formulas of Sec. IIC remain valid.

## 3 Spin systems

### 3.1 Erbium-doped CaWO$_4$

The first system investigated consists of erbium $Er^{3+}$ ions doped into a $CaWO_4$ matrix, substituting $Ca^{2+}$. The crystal has a tetragonal body-centered structure (see Fig. 3) with lattice constants $a = b = 0.524$ nm and $c = 1.137$ nm. Rare-earth ions with an odd number of electrons such as $Er^{3+}$ have a ground state consisting of two levels that are degenerate in zero magnetic field, and separated from other levels by an energy scale equivalent to several tens of Kelvin due to the crystalline electric field and the spin-orbit interaction. This pair of electronic levels is known as a Kramers doublet, and forms an effective $S_0 = 1/2$ electron spin system, with a spin Hamiltonian $H_{Er}$ (Abragam and Bleaney, 2012) whose form is given by Eq.(12).

Due to the S4 site symmetry in which rare earth ions are found in $CaWO_4$, the g-tensor is diagonal in the crystallographic frame with $g_{xx} = g_{yy} = 8.38$ and $g_{zz} = 1.247$ (Antipin et al., 1968) ($x, y, z$ corresponding to $a, b, c$). Of all erbium atoms, 77% are from an isotope that has nuclear spin $I_0 = 0$ and therefore no contribution from the hyperfine term in Eq.(12). Their energy levels are shown in Fig.3 for $B_0$ applied in the $(a, b)$ plane.

The remaining 23% are from the $^{167}Er$ isotope with $I_0 = 7/2$. Its hyperfine coupling tensor to the $Er^{3+}$ electron spin is diagonal, with coefficients $A_{xx} = A_{yy} = 873$ MHz and $A_{zz} = 130$ MHz. The 16 eigenfrequencies of the $^{167}Er$ spin Hamiltonian are also shown in Fig.3, again for $B_0$ applied in the $(a, b)$ plane. In the high-magnetic field limit $B_0 \gg A_{Er}/(g_{Er}\beta_e)$, which is satisfied in the measurements reported below, the eigenstates are simply described by $|\pm, m_I\rangle$, $\pm$ describing the electron spin quantum number $m_S = \pm 1/2$ and $m_I$ the nuclear spin quantum number. The EPR-allowed transitions are the transitions between levels that preserve $m_I$. Therefore we can apply the fictitious spin model with $|\alpha, \beta\rangle = |\pm, m_I\rangle$.

The $CaWO_4$ matrix also contains nuclear spins. Indeed, the $^{183}W$ isotope has a spin $I = 1/2$ with nuclear g-factor $g_n = 0.235$ (corresponding to a gyromagnetic ratio of 1.8 MHz/T), and is present in a $p = 0.13$ abundance, whereas the other tungsten isotopes are nuclear-spin-free. The interaction of the $^{183}W$ atoms with the erbium ions gives rise to the ESEEM studied below. Because the 4f electron wavefunction is mainly located on the $Er^{3+}$ ion, the contact hyperfine with the nuclear spins of the lattice is expected to be negligibly small. We therefore model the hyperfine interaction with $^{183}W$ by the dipole-dipole term in Eq.(14).

### 3.2 Bismuth donors in Silicon

The other system considered is the bismuth donor in silicon. Bismuth, as an element of the 5th column, substitutes in the silicon lattice by making 4 covalent bonds with neighboring atoms, leaving one unpaired electron that can be weakly trapped by the hydrogenic potential generated by the $Bi^+$ ion, whose spin gives rise to the resonance signal (see Fig.4a). The donor wavefunction $\psi(\boldsymbol{r})$ has a complex structure that extends over $\approx 1.5$ nm in the silicon lattice (Kohn and Luttinger, 1955; Feher, 1959) (see Supp. Info). As for $Er : CaWO_4$, the donor spin Hamiltonian $H_{Bi}$ is given by Eq.(12). However in this case the g-tensor $g_e \mathbf{1}$ is isotropic with $g_e = 2$, and the hyperfine tensor $A_{Bi}\mathbf{1}$ with the nuclear spin $I_0 = 9/2$ of the Bismuth atom is also isotropic, with $A_{Bi}/2\pi = 1.4754$ GHz.

The eigenstates of $H_{Bi}$ have simple properties because of its isotropic character. Denoting $m_S$ ($m_I$) the eigenvalue of $S_{z,0}$ ($I_{z,0}$), we note that $m = m_I + m_S$ is a good quantum number since $H_{Bi}$ commutes with $S_{z,0} + I_{z,0}$ (Mohammady et al., 2010), $z$ being the direction of $\boldsymbol{B_0}$. States with equal $m$ are hybridized by $H_{Bi}$. States $|m = 5\rangle$ and $|m = -5\rangle$, corresponding to $|m_S = +1/2, m_I = 9/2\rangle$ and $|m_S = -1/2, m_I = -9/2\rangle$, are non-degenerate and are thus also eigenstates of $H_{Bi}$. States with $|m| \leq 4$ belong to 9 two-dimensional subspaces spanned by $|m_S = +1/2, m_I = m - 1/2\rangle, |m_S = -1/2, m_I = m + 1/2\rangle$ within which the 2 eigenstates of $H_{Bi}$ are given by $|\pm, m\rangle = a_m^\pm |\pm \frac{1}{2}, m \mp \frac{1}{2}\rangle + b_m^\pm |\mp \frac{1}{2}, m \pm \frac{1}{2}\rangle$, with values of $a_m^\pm, b_m^\pm$ that can be determined analytically (Mohammady et al., 2010).



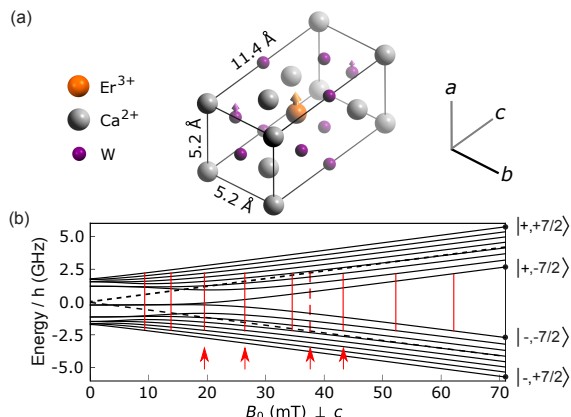

**Figure 3.** Structure and energy diagram of erbium ions in $CaWO_4$. (a) Crystal structure with oxygen atoms hidden for clarity. Erbium atoms are in substitution of the Calcium. The crystal has a rotational symmetry around the $c$ axis. A fraction $p = 0.13$ of the $W$ atoms are of the $^{183}W$ isotope, with a nuclear spin $1/2$. (b) Energy level diagram of the $I = 0$ erbium isotopes (black dashed line) and of the $^{167}Er$ isotope (black solid lines) with $I = 7/2$, for $B_0$ applied perpendicular to the $c$ axis. Red vertical lines indicate the value of $B_0$ for which an allowed EPR transition becomes resonant with the $4.372\,GHz$ frequency of our detection resonator (see text Sec.IV). Four red arrows indicate the values of $B_0$ at which ESEEM data were measured.

Contrary to the erbium case, the measurements of bismuth donor spins are performed in the low-field limit $|g_e\beta_e B_0| \ll |A_{Bi}|$, in which the eigenstates are fully hybridized. In this limit, a useful approximate expression for the eigenenergy of level $|\pm, m\rangle$ is

$$E_m^\pm \approx -\frac{A_{Bi}}{2} \pm \frac{5A_{Bi}}{2} \pm \frac{mg_e\beta_e B_0}{10}. \tag{16}$$

The magnetic-field dependence of the $|\pm, m\rangle$ energy levels is shown in Fig. 4(b) for $B_0 < 1\,mT$. Note in particular that the separation between neighboring hyperfine levels is given by $E_m^\pm - E_{m-1}^\pm \approx +\frac{g_e\beta_e B_0}{10} = \pm 2\pi \times 2.8 B_0\,GHz$.

    Because of the hybridization, all transitions that satisfy $|\Delta m| = 1$ are to some extent EPR-allowed at low field i.e., have a non-zero matrix element of operator $S_{0,x}$. In this work, we particularly focus on the $18\ |\Delta m| = 1$ transitions that are in the $\simeq 7\,GHz$ frequency range at low magnetic fields $|+, m\rangle \leftrightarrow |-, m-1\rangle$ and $|-, m\rangle \leftrightarrow |+, m-1\rangle$, as shown in Fig.4c. The
$|-, m\rangle \leftrightarrow |+, m+1\rangle$ and $|-, m+1\rangle \leftrightarrow |+, m-1\rangle$ transitions are degenerate in frequency for $-4 \leq m < 4$ as seen from Eq.(16), which results in only 10 different transition frequencies (see Figs. 4b,c, and 8a).

    The most abundant isotope of silicon is $^{28}Si$, which is nuclear-spin-free. The lattice also contains a small percentage $p$ of $^{29}Si$ atoms that have a nuclear spin $I = 1/2$ and give rise to the ESEEM. The g-factor of $^{29}Si$ is $g_n = -1.11$, yielding a gyromagnetic ratio of $8.46\,MHz/T$.
The donor-$^{29}Si$ hyperfine interaction is given by Eq.(14). Due to the spatial extent of the electron wavefunction, the Fermi contact term is not negligible and needs to be taken into account together with the dipole-dipole coupling (Hale and Mieher, 1969); more details can be found in the Supplementary Information.

    The restriction of the total system Hamiltonian to each of the 18 ESR-allowed transitions of the Bismuth donor manifold can be mapped onto the fictitious spin-1/2 model of Section IID. Note however that the hyperfine term $|A_j|$ can take values up to
$\sim 1\,MHz$ for proximal nuclear spins, which is comparable to or larger than the frequency difference between hyperfine states of the Bismuth donor manifold at low field as explained above. The validity of the fictitious spin-1/2 model in this context will be discussed in Section VI.

## 4   Experimental setup and samples

    The EPR spectrometer has been described in detail in refs. (Bienfait et al., 2015; Probst et al., 2017) and is shown schematically
in Fig.5a. It is built around a superconducting micro-resonator consisting of a planar interdigitated capacitor shunted by an





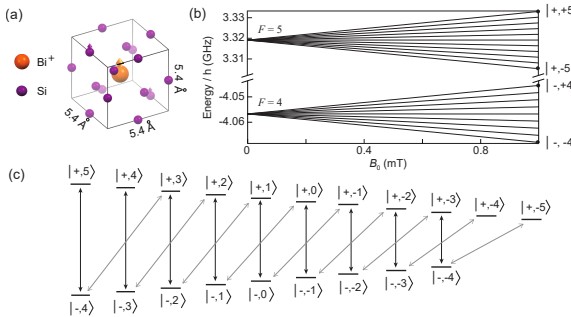

**Figure 4.** Structure and energy diagram of bismuth donors in silicon. (a) Silicon crystal structure, showing a substitutional bismuth atom coupled to nearby $^{29}$Si nuclear spins. The donor electron is trapped around the Bi$^+$ ion and its wavefunction covers many lattice sites. (b) Energy levels of the bismuth donor, for $B_0 < 1$ mT. (c) Schematic representation of the allowed transitions (black and grey arrows) between the bismuth donor energy levels in the low field limit.

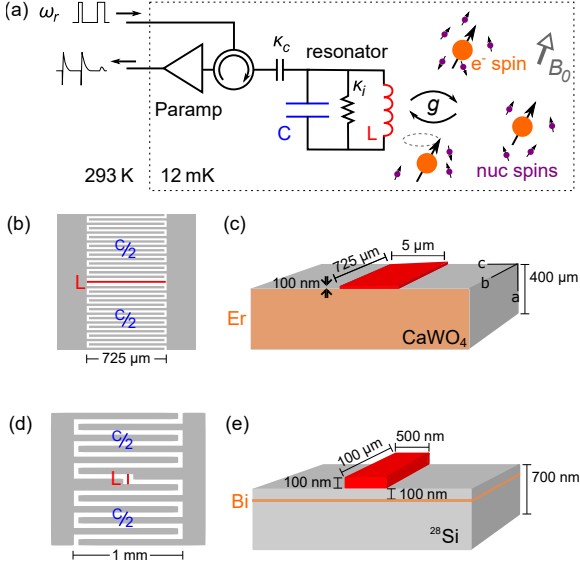

**Figure 5.** Experimental setup and samples. (a) Schematic of the low-temperature EPR spectrometer. The LC resonator is inductively coupled to electron spins, which are coupled to a nuclear spin bath that causes the ESEEM. The spins are probed by sequences of microwave pulses at the resonator frequency $\omega_r = 1/\sqrt{LC}$. Reflected pulses as well as the echo signal are routed to a parametric amplifier, and are further amplified at 4 K, and finally demodulated and digitized at room temperature. (b,c) Design of the LC resonator used for the detection of erbium ion spins, with a 725$\mu$m-long, 5$\mu$m-wide inductor. It is patterned out of a 100nm thick niobium film deposited on top of a CaWO$_4$ substrate bulk-doped with Er$^{3+}$ ions. (d,e) Design of the LC resonator used for the detection of bismuth donor spins, with a 100$\mu$m-long, 0.5$\mu$m-wide inductor. It is patterned out of a 100nm thick aluminum film deposited on top of a silicon substrate isotopically enriched in $^{28}$Si, in which bismuth ions were implanted at a 50-100nm depth.

inductor, directly patterned on the crystal. We detect the spins that are located in the immediate vicinity of the resonator inductance. Note that the microwave $B_1$ field generated by the inductance is spatially inhomogeneous. If the spin location is broadly distributed, this can make the application of control pulses with a well-defined Rabi angle problematic(Ranjan et al., 2020). As explained below, the resonator is more strongly coupled to the measurement line than in Ref. (Bienfait et al., 2015) to increase the measurement bandwidth as requested for ESEEM spectroscopy.





The sample is mounted in a copper sample holder thermally anchored at the mixing chamber of a dilution refrigerator. A DC magnetic field $B_0$ is applied parallel to the sample surface and along the resonator inductance. Microwave pulses for spin coherent excitation are sent to the resonator input through a heavily attenuated line, and their reflection or transmission, together with the echo signal emitted by the spins, is fed into a superconducting Josephson Parametric Amplifier, either of the flux-pumped type (Zhou et al., 2014) or of the Josephson Traveling-Wave Parametric Amplifier (JTWPA) type (Macklin et al., 2015). Further microwave amplification takes place at 4K and room-temperature, before homodyne demodulation which yields the two signal quadratures $[I(t), Q(t)]$. The echo-containing quadrature signal is then integrated to yield the echo amplitude $A_e$. Such a setup was shown to reach sensitivities of order $10^2 - 10^3$ spin/$\sqrt{Hz}$ (Bienfait et al., 2015; Eichler et al., 2017; Probst et al., 2017).

The erbium-doped sample (from Scientific Materials) was prepared by mixing erbium oxide with calcium and tungsten oxides before crystal growth, yielding a uniform Er concentration of $6 \cdot 10^{17}$ cm$^{-3}$ (50 ppm) throughout the sample. For resonator fabrication, the bulk crystal was cut and polished to a thin rectangular sample with dimensions $0.4$mm $\times 3$mm $\times 6$mm parallel to $a \times b \times c$ axes. The resonator was patterned out of a 100 nm thick (sputtered) Nb layer, using a design similar to that shown in Ref (Bienfait et al., 2015). More specifically, 15 interdigitated fingers on either side of a $720 \mu$m $\times 5 \mu$m inductive wire form an LC resonator, corresponding to a detection volume of $V_{Er} \sim 20$ pL. In the absence of magnetic field, the resonance frequency is $\omega_r/2\pi = 4.323$ GHz. Its total quality factor of $8 \cdot 10^3$ is set both by the internal losses, characterized by the energy loss rate $\kappa_i = 5 \cdot 10^5$s$^{-1}$, and by its coupling to the measurement line $\kappa_C = 3 \cdot 10^6$s$^{-1}$.

The bismuth donors have been implanted at $\approx 100$ nm depth with a peak concentration of $8 \cdot 10^{16}$ cm$^{-3}$ in a silicon sample. They lie in a 700 nm-thick silicon epilayer enriched in the nuclear-spin-free $^{28}$Si isotope (nominal concentration of 99.95%), grown on top of a natural-abundance silicon sample. The resonator is patterned out of a 50nm-thick aluminum film. It has the same geometry as reported in (Probst et al., 2017), with a 100 $\mu$m-long, 500 nm-wide inductor, and a detection volume of 0.2 pL. Its frequency $\omega_r/2\pi = 7.370$ GHz is only slightly below the zero-field splitting of unperturbed Bi:Si donors $5A_{Bi}/(2\pi) = 7.37585$ GHz (Wolfowicz et al., 2013). The resonator internal loss is given by $\kappa_i = 3 \cdot 10^5$ s$^{-1}$. The coupling to the measurement line can be tuned at will by modifying the length of a microwave antenna that capacitively couples the measurement waveguide to the on-chip resonator via the copper sample holder (Bienfait et al., 2015; Probst et al., 2017). For the experiments reported below we used two settings : one for which the resonator was over-coupled ($\kappa_{C1} = 10^7$ s$^{-1}$), corresponding to a loaded quality factor $Q_1 = 4 \cdot 10^3$, and one for which the coupling was closer to critical ($\kappa_{C2} = 10^6$ s$^{-1}$), corresponding to a loaded quality factor $Q_2 = 3.4 \cdot 10^4$. In the low-Q case, square microwave pulses were used, of duration $\simeq 100$ ns similar to the cavity field damping time. In the high-Q case, shaped pulses were used (Probst et al., 2019) so that the intra-cavity field was a square pulse of 1 $\mu$s without any ringing. In some experiments, we additionally used a train of $\pi$ pulses (CPMG sequence), which generated extra echoes for significant gain in signal-to-noise ratio. More details on the pulse sequences used, the phase cycling scheme, and the repetition time, will be given in the following sections, together with experimental results.

## 5   Results

### 5.1   Erbium-doped CaWO$_4$

#### 5.1.1   Spectroscopy

Figure 6 shows a spectrum comprising a series of microwave transmission measurements recorded on a vector network analyser, measured at 100 mK, as a function of the magnetic field $B_0$ applied along the $b$ crystal axis (Probst et al., 2020). Note that compared to Fig.5a, the resonator is coupled to the measurement line in a hanger geometry (Day et al., 2003), so that its resonance appears as a dip in the amplitude transmission coefficient $|S_{21}|$ (see Fig.6. The 9 red lines indicate the values of $B_0$ at which the calculated Er$^{3+}$ ion transitions are equal to $\omega_r$ (see Fig. 3b). Avoided level crossings are observed, which indicate a strong coupling of the resonator to the erbium transitions. Several additional anti-crossings and discontinuities are visible above 40mT. These are attributed to ytterbium impurities $\left(^{171}Yb \text{ and } ^{173}Yb\right)$ and magnetic flux vortices penetrating the resonator.

Noticeable in the spectrum at 37mT is the large anti-crossing attributed to the highly concentrated $I = 0$ erbium isotopes. Here the high-cooperativity regime ($C > 30$) is reached between the electronic spins and the resonator (Kubo et al., 2010; Probst et al., 2013). Typical linewidths $\Gamma/2\pi \sim 20$ MHz is observed. The coupling strength is also observed to be different for the eight $^{167}$Er transitions, which are labeled according to their corresponding nuclear spin projections $m_I$. This is explained by the partial polarisation of the ground-state hyperfine levels of $^{167}$Er$^{3+}$ at millikelvin temperatures (see Fig. 3b).





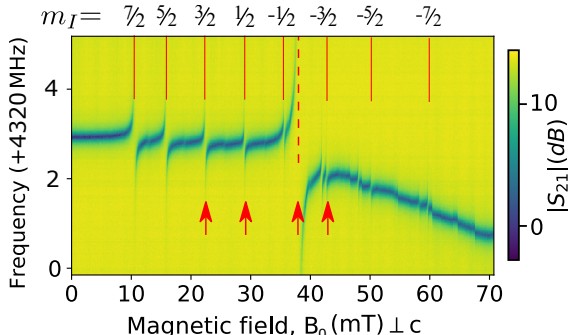

**Figure 6.** Spectroscopy of $\mathrm{Er}^{3+}$:$\mathrm{CaWO_4}$. Transmission coefficient $|S_{21}|(\omega)$ at $100\,\mathrm{mK}$ as a function of the magnetic field $B_0$ applied along to the $a$ crystalline axis, around $4.323\,\mathrm{GHz}$. Red vertical lines indicate the expected Erbium transitions either for the $I=0$ isotopes (dashed) or the $I=7/2$ isotope (solid). Red arrows indicate the field at which the ESEEM data are measured.

### 5.1.2 Two-Pulse ESEEM

Four values of $B_0$ were selected for investigating ESEEM, indicated by the arrows in Fig. 6; the first, second, and fourth corresponding to electronic-spin transitions of $^{167}\mathrm{Er}$, and the third one to the $I=0$ isotopes. The two-pulse echo sequence of Fig.1a was implemented with square pulses of $1\mu$s duration applied at the resonator input, with double amplitude for the second pulse. Note that due to the $B_1$ spatial inhomogeneity combined with the homogeneous spin distribution throughout the crystal, the spread of Rabi frequency is too large to observe a well-defined nutation signal. The Rabi angle is therefore not well defined, and the echo is the average of different rotation angles.

The control pulses driving the spins are filtered by the resonator bandwidth $\kappa/2\pi \simeq 600\,\mathrm{kHz}$, corresponding to a field decay time $2\kappa^{-1} = 3.3\mu$s. The repetition time between echo sequences was 1 second, close to the measured spin relaxation time. The echo signal was averaged 10 times with phase-cycling of the $\pi$-pulse to improve signal-to-noise and to remove signal offsets.

Figure 7 shows the two-pulse echo integrated amplitude $A_e$ as a function of $\tau$ for each of the four Er transitions investigated (Probst et al., 2020). A clear envelope modulation signal is observed, together with an overall damping. Here we are interested only in the modulation pattern; a detailed study of the coherence time $T_2$ will be provided elsewhere. Qualitatively, we observe that the modulation frequency increases with $B_0$ and the modulation amplitude overall decreases with $B_0$, as expected from the discussion in Section II. A Fourier transform of the $I=0$ data (see Fig. 7b) shows the ESEEM spectrum. Well resolved peaks are observed in the $5-100\,\mathrm{kHz}$ range, distributed around the $^{183}\mathrm{W}$ bare Larmor frequency $\omega_\mathrm{W}$.

A very rough estimate of the number of erbium ions contributing to the signal is $[\mathrm{Er}]V_\mathrm{Er}\kappa/\Gamma$, which is $2.5\cdot 10^8$ for the $I=0$ data, and $10^7$ for each $^{167}\mathrm{Er}$ transition.

### 5.1.3 Comparison with the model

We compute the echo envelope $V'_\mathrm{2p}(\tau)$ described in Section II, with the nearest 1000 coupled tungsten nuclei ($N=1000$) and a natural $^{183}\mathrm{W}$ abundance of $14.4\%$ ($p=0.144$). The hyperfine interaction is taken to be purely dipolar, as already explained (Guillot-Noël et al., 2007; Car et al., 2018). The fitting proceeds by assigning an initial 'guess' to six free parameters, then minimising using the L-BFGS-B algorithm (Byrd et al., 1995). Three of these parameters ($|B_0|,\phi,\theta$) describe the applied magnetic field:

$$B_0 = |B_0|\left[\sin\theta\cos\phi\,\hat{x} + \sin\theta\sin\phi\,\hat{y} + \cos\theta\,\hat{z}\right]$$

Here $\theta$ is the angle of the field relative to the crystal $c$-axis ($\hat{z}$) and $\phi$ is the angle relative to the $a$-axis ($\hat{x}$) in the $a$-$b$ plane ($\hat{x}$-$\hat{y}$ plane). The other three parameters ($C,T_2,n$) account for the echo envelope decay

$$\mathrm{A_e}(\tau) = V_\mathrm{2p}(\tau)\cdot C\exp\left(-\frac{2\tau}{T_2}\right)^n,$$





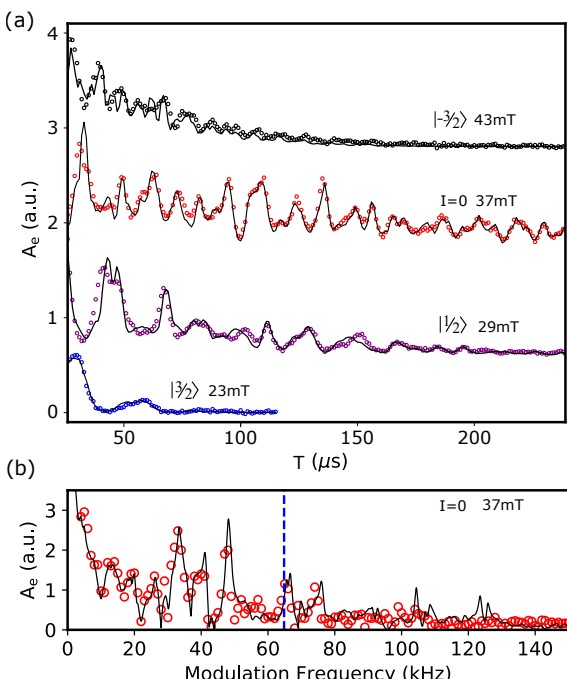

**Figure 7.** Two-pulse ESEEM on Er:CaWO4. (a) Integrated echo area $A_e$ as a function of the inter-pulse delay $\tau$, for 4 values of $B_0$ corresponding to different transitions. Open circles are measurements, and solid lines are the results of the ESEEM calculations as explained in the text Sec. V.A.3. (b) Measured (open red circles) and computed (solid line) fast Fourier transform of the $I = 0$ data. The blue dashed line shows the Larmor frequency of $^{183}$W nuclei in free space.

where $C$ represents the signal magnitude, $T_2$ the coherence time and $n \in [1, 2]$ accounts for non-exponential decay. To determine the global minimum of the fit, the minimisation is repeated 200 times with randomly seeded initial values for the six parameters, bounded within the known uncertainty of the applied magnetic field $B_0$, signal strength $C$ and coherence time $T_2$. This approach reveals single local minima for each fitted parameter within the bounded range, with the variance of the 200

outcomes determining the uncertainty for each parameter. In particular, it yields precise values for the angles $\theta = 91.47 \pm 0.01^0$ and $\phi = 90.50 \pm 0.01^0$. The result of this fitting is presented in Fig.7(a), overlaid on the data for the $I = 0$ transition at 37mT. Only the decay parameters $(C, T_2, n)$ and magnetic field magnitude $|B_0|$ are left free when fitting the other three transitions in Fig. 7(a). This was done for consistency between data sets, and because the $I = 0$ data yields the most accurate values for $\phi$ and $\theta$ due to the low decoherence rate.

Note that good fits to the data are also achieved by including only the nearest 100 tungsten nuclei, although noticeable deviations between the data and fit are observed with any less. The dimensionless 'anisotropic hyperfine interaction parameter' $\rho$ described in the seminal publication on ESEEM (Rowan et al., 1965) is not required here. This parameter was introduced with the earliest attempts of ESEEM fitting, likely to compensate for the low number of simulated nuclear spins (typically 10 nearest nuclei or less), and was interpreted as an account for a potential distortion of the local environment caused by dopant insertion.

Finally, a consideration of the spectral components presented in Fig.7(b) helps to more clearly identify the difference between the fit and the data. In particular, the high frequency components of the fitted model are not present experimentally due to the filtering effect of the superconducting resonance (260 kHz HWHM). This high-Q resonator greatly reduces the bandwidth of the RF field absorbed by the coupled Er-$^{183}$W system and further limits the bandwidth of the detected echo signal.





## 5.2 Bismuth donors sample

### 5.2.1 Spectroscopy

Given the resonator frequency $\omega_r$, four bismuth donor resonances should be observed when varying $B_0$ between $0$ and $1\,\mathrm{mT}$, as seen in Fig.8a. Figure 8(b) shows an echo-detected field sweep, measured at $12\,\mathrm{mK}$: the integrated area $A_e$ of echoes obtained with a sequence shown in Fig. 1a with $\tau = 50\,\mu\mathrm{s}$ pulse separation is plotted as a function of $B_0$ (Probst et al., 2020). Instead of showing well-separated peaks as in the Erbium case, echoes are observed for all fields below $1\,\mathrm{mT}$, with a maximum close to $0.1\,\mathrm{mT}$, and extends in particular down to $B_0 = 0\,\mathrm{mT}$. This is the sign that each of the expected peaks is broadened and overlaps with neighboring transitions. Close to zero field, the echo amplitude goes down by a factor 2 on a scale of $\sim 0.1\,\mathrm{mT}$, before showing a sharp increase at exactly zero field. These zero-field features are not currently understood, but they are reproducible as confirmed by the measurements at $B_0 < 0$, which are approximately symmetric to the $B_0 > 0$ data as they should be.

Line broadening was reported previously for bismuth donors in silicon in related experiments (Bienfait et al., 2015; Probst et al., 2017), and was attributed to the mechanical strain exerted by the aluminum resonator onto the silicon substrate due to differential thermal contractions between the metal and the substrate. At low strain, $A_{\mathrm{Bi}}$ depends linearly on the hydrostatic component of the strain tensor $\epsilon_{\mathrm{hs}} = (\epsilon_{xx} + \epsilon_{yy} + \epsilon_{zz})/3$ with a coefficient $dA_{\mathrm{Bi}}/d\epsilon_{\mathrm{hs}}/(2\pi) = 28$ GHz(Mansir et al., 2018). Quantitative understanding of the lineshape was achieved in a given sample geometry based on this mechanism (Pla et al., 2018), using a finite-element modelling to estimate the strain profile induced upon sample cooldown. A similar modelling was performed for the Bi sample reported here (see Fig. 8(d)). Based on the typical strain distribution $|\epsilon_{hyd}| \sim 3 \cdot 10^{-4}$ and on the hyperfine to strain coefficient $dA_{\mathrm{Bi}}/d\epsilon_{\mathrm{hs}}/(2\pi) = 28$ GHz, we expect the zero-field splitting $5A_{\mathrm{Bi}}/(2\pi)$ to have a spread of $\sim 50$MHz, which would indeed result in complete peak overlap in the $B_0 < 1\,\mathrm{mT}$ region, as observed in Fig. 8(b).

This broadening has two consequences worth highlighting. First, the bismuth donor echo signals can be measured down to $B_0 = 0$mT, which otherwise is generally impossible in X-band spectroscopy. Here, this is enabled by the large hyperfine coupling of the Bi:Si donor, combined with strain-induced broadening. This makes it possible to detect ESEEM caused by very-weakly-coupled nuclear spins, which requires low magnetic fields as explained in Section II. Second, at a given magnetic field, the spin-echo signal contains contributions from several overlapping EPR transitions. This last point is best understood from Fig. 8(c), which shows how several classes of Bismuth donors, each with different hyperfine coupling $A_{\mathrm{Bi}}$, may have transitions resonant with $\omega_r$. We will assume in the following that the inhomogeneous distribution of $A_{\mathrm{Bi}}$ is so broad that each of the 10 $A_{\mathrm{Bi}}$ values for which one bismuth donor transition is resonant with $\omega_r$ at fixed $B_0$ is equally probable, which is likely to be valid for $B_0 < 1$mT.

### 5.2.2 Two-Pulse ESEEM

Two-pulse echoes are measured with the pulse sequence shown in Fig. 1, which consists of a square $\pi/2_{\mathrm{X}}$ pulse of duration $50$ ns followed by a square $\pi_{\mathrm{Y}}$ pulse of duration $100$ ns after a delay $\tau$. Note that due to the donor spatial location in a shallow layer below the surface and to the strain shifting of their Larmor frequency (Pla et al., 2018), the Rabi frequency is more homogeneous than in the erbium-doped sample, and Rabi rotations with a well-defined angle can be applied (Pla et al., 2018; Probst et al., 2017). To increase the signal-to-noise ratio, a CPMG sequence of 198 $\pi$ pulses separated by $10\,\mu s$ are used following the echo sequence (Probst et al., 2017). The curves are repeated 20 times, with a delay of $2$ s in-between to enable spin relaxation of the donors. All the resulting echoes are then averaged. Phase cycling is performed by alternating sequences with opposite phases for the $\pi/2$ pulses and subtracting the resulting echoes. The data are obtained in the low-Q configuration (see section IV).

Figure 9 shows the integral of the averaged echoes $A_e(\tau)$ as a function of $\tau$, for various values of $B_0$ (Probst et al., 2020). At non-zero field, $A_e(\tau)$ shows $B_0$-dependent oscillations on top of an exponential decay with time constant $T_2 = 2.6$ ms. Similar decay times were measured on the same chip with another resonator (Probst et al., 2017), and are attributed to a combination of donor-donor dipolar interactions and magnetic noise from defects at the sample surface.

In the subsequent discussion, we concentrate on the ESEEM pattern. To analyze the data, each curve was divided by a constant exponential decay with 2.6 ms time constant, mirrored at $t = 0$, and Fourier transformed (see Fig. 10). Only two peaks are observed. Their frequencies vary linearly with $B_0$, and are found to be approximately $8$ kHz/mT and $16$ kHz/mT. This is in good agreement with the gyromagnetic ratio of $^{29}$Si ($8.46$kHz/mT); the presence of the second peak at twice this value is expected as explained in Section II for the two-pulse ESEEM in the weak-coupling limit. The oscillation amplitude goes down with $B_0$, again as expected from the model put forward in Section II.

A rough estimate of the number of donors contributing to the measurements shown in Fig. 9 can be obtained by comparison with (Probst et al., 2017). Given the nearly identical resonator geometry, and assuming identical strain broadening in both samples, the ratio of the number of donors involved in both measurements is simply given by the ratio of resonator bandwidths. For





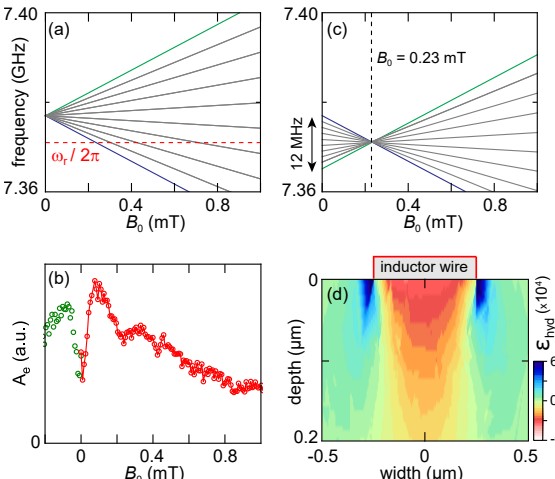

**Figure 8.** (a) EPR-allowed transitions of a bismuth donor in silicon for $0 < B_0 < 1$ mT. The red dashed line denotes the resonator frequency $\omega_r$. The spectrum is for an unstrained donor, for which the frequency at $B_0 = 0$ is $5A_{Bi}/(2\pi)$. (b) Echo-detected field sweep. The echo integral $A_e$ is plotted versus $B_0$. (c) Frequency of all 18 Bismuth donor transitions that may contribute to the echo signal at a given field (here, $B_0 = 0.23$ mT). This is made possible by the strain-induced spread in $A_{Bi}$ between different donors. (d) Hydrostatic component of strain in silicon simulated using COMSOL.

the low-Q configuration, such as the two-pulse-echo of Fig. 9, this corresponds to $\simeq 5 \cdot 10^3$ dopants; in the high-Q configuration (see the 3- and 5-pulse data in the next paragraph), this number is reduced to $\simeq 5 \cdot 10^2$ dopants.

### 5.2.3   Three- and Five-Pulse ESEEM

The spectral resolution provided by the measurement protocol is limited because of the finite electron coherence time $T_2$. As
discussed in Section II, this can be overcome by 3- or 5-pulse ESEEM.

We measure 3- and 5- pulse ESEEM with the pulse sequence shown in Fig. 11. The high-Q configuration is chosen; shaped pulses generate an intra-cavity field in the form of a rectangular pulse of $1\ \mu$s duration with sharp rise and fall (Probst et al., 2019) despite the high resonator quality factor. The data are acquired at $B_0 = 0.1$ mT, so that $\omega_I/2\pi \simeq 850$ Hz. The first blind spot for 3-pulse ESEEM is thus at $2\pi/\omega_I = 1.2$ ms; we chose $\tau = 290\ \mu$s for the 3-pulse echo, and $\tau_1 = \tau_2 = 290\ \mu$s for the
10 5-pulse sequence. A sequence of 19 CPMG $\pi$ pulses, separated by $50\ \mu$s, was used to enhance the signal-to-noise ratio. The sequences were repeated after a fixed waiting time of 100 ms between the last $\pi$ pulse of one sequence and the first $\pi/2$ pulse of the following, to enable spin relaxation. Phase-cycling is used to suppress unwanted echoes (see Supplementary Information for the schemes (Schweiger and Jeschke, 2001; Kasumaj and Stoll, 2008)). Each point is averaged over $2.5 \cdot 10^4$ sequences, with a total acquisition time of 2 weeks for each curve (Probst et al., 2020).
The results are shown in Fig. 11, together with their fast Fourier transform (Probst et al., 2020). Both the 3-pulse ESEEM (3PE) and 5-pulse ESEEM (5PE) curves show oscillations that last one order of magnitude longer than the electron spin $T_2$ (up to 20 ms), enabling higher spectral resolution of the ESEEM signal. The 5PE curve has a higher oscillation amplitude than the 3PE by a factor 2-3, as expected. The decay of the oscillations occurs in $\sim 10$ ms, one order of magnitude faster than the stimulated echo amplitude (see the 3PE curve), suggesting that it is an intrinsic feature of the ESEEM signal, as discussed
below.

The spectrum shows only one peak at the $^{29}$Si frequency. This is consistent with the expression provided in Sec. II and the Supplementary Information for the 3- and 5-pulse ESEEM, in which the terms oscillating at the sum and difference frequency are absent in contrast to the 2-pulse ESEEM. The peak width is $\simeq 100$ Hz, which indicates that the nuclei contributing to the ESEEM signal have hyperfine coupling strengths $A, B$ of at most 100 Hz. Neglecting the contact interaction term, this
corresponds to $^{29}$Si nuclei that are located at least $\sim 5$ nm away from the donor spin.

The measured ESEEM spectrum of the bismuth donor sample qualitatively differs from the erbium sample, since it only contains a peak at the unperturbed silicon nuclei Larmor frequency (and at twice this frequency for the 2-pulse ESEEM),



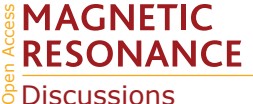

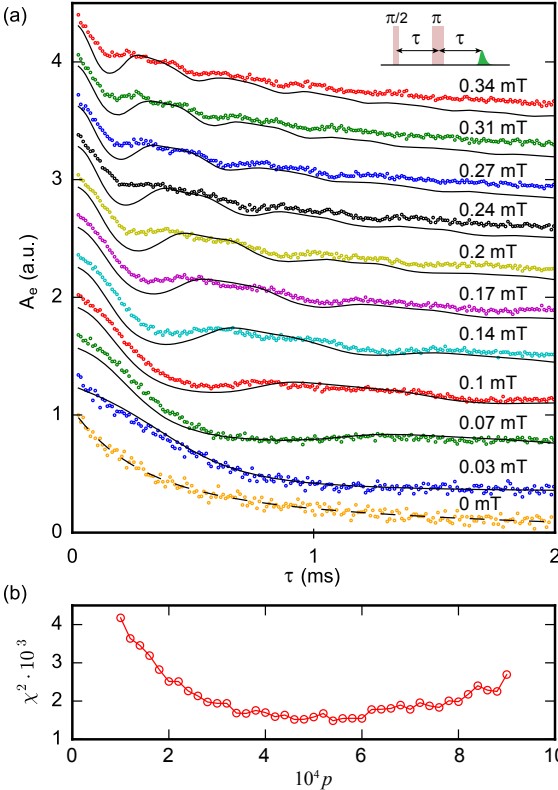

**Figure 9.** Two-pulse ESEEM: (a) Echo integral $A_e$ versus inter-pulse delay $\tau$ for a 2-pulse echo sequence, for varying magnetic field $B_0$. Dots are experimental data, lines are results of the model (see text), assuming a concentration in $^{29}$Si of $p = 4.4 \cdot 10^{-4}$. The curves are vertically shifted, for clarity (b) Fit residue $\chi^2$ for different $^{29}$Si relative abundance $p$. The best fit is obtained for $p = 4.4 \pm 1 \cdot 10^{-4}$, in agreement with the specified value.

instead of the many peaks observed in Fig.7 indicating nuclear spin contribution with vastly different hyperfine strengths. This can be qualitatively understood by examining Eq.10. Defining $N_l$ as the number of lattice sites with approximately the same hyperfine parameters $A_l, B_l$ and modulation frequency $\omega_{\downarrow/\uparrow,l}$, the component at $\omega_{\downarrow/\uparrow,l}$ is visible in the spectrum if $N_l k_l p \sim 1$, which can only be achieved if $N_l p \sim 1$. In the case of erbium, $p = 0.144$ so that even the sites closest to the ion (for which $N_l$ is of order unity) may satisfy this condition for well-chosen $B_0$. In the bismuth donor sample where $p = 4.4 \cdot 10^{-4}$, this condition can only be met for $N_l \sim 10^3$, and therefore for crystal sites $l$ that are far from the donor, for which the hyperfine coupling is small, so that $\omega_{\downarrow/\uparrow,l} \simeq \omega_I$. This is confirmed by the more quantitative modelling below.

### 5.2.4 Comparison with the model

As explained above, the measured echo signal results from the contribution of all 18 Bi:Si transitions because of strain broadening. To model the data, we therefore apply the fictitious spin-1/2 model to each transition, and sum the resulting echo amplitudes weighted by their relative contribution, which we determine using numerical simulations described in the Supplementary Information.

Moreover, as discussed in Section III, and in contrast to the erbium case, the fictitious spin model for a given transition needs to be validated in the low-$B_0$ regime because the energy difference between neighboring hyperfine levels of the bismuth donor manifold $(E_m^\pm - E_{m-1}^\pm)/h \simeq 0.3\,\text{MHz}$ for $B_0 = 0.1\,\text{mT}$ is comparable to or even lower than the hyperfine coupling to some $^{29}$Si nuclei. In that case, the hyperfine interaction induces significant mixing between the bismuth donor and the $^{29}$Si



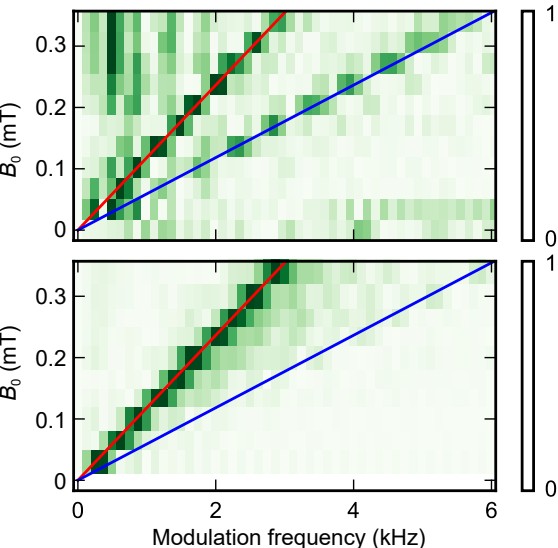

**Figure 10.** Amplitude of the Fourier transform of the experimental (top panel) and theoretical (bottom panel) 2-pulse ESEEM data.

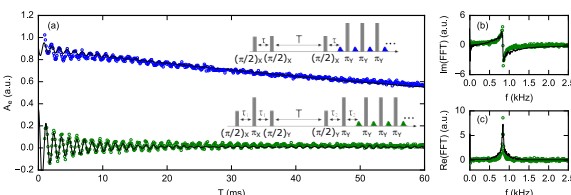

**Figure 11.** (a) 3-pulse (blue circles) and 5-pulse (green circles) ESEEM signals at $B_0 = 0.1$ mT. Black lines are simulations assuming a $^{29}$Si concentration $p = 4.4 \cdot 10^{-4}$. (b) Imaginary and (c) real part of the Fourier transform of the 5-pulse ESEEM data. The spectrum only contains a peak at $850$ Hz, which is the $^{29}$Si nuclei Larmor frequency at this field.

eigenstates, and we should describe the coupled electron spin $\mathbf{S}_0$- $^{209}$Bi nuclear spin-$\mathbf{I}_0$+$^{29}$Si nuclear spin $\mathbf{I}$ as a single 40-level quantum system.

This study is described in the Supplementary Information Sec.IV for a $^{29}$Si with strong hyperfine coupling ($\geq 200$ kHz). The state mixing makes many transitions EPR-allowed, and the interference between these transitions causes fast oscillations in the spin echo signal, as seen in Fig. S7 in the Supplementary Information. The frequencies of these oscillations depend greatly on the local Overhauser field on the donor electron spin. Since the latter has a large inhomogeneous broadening ($\sim 0.5$ MHz), the ensemble average leads to a rapid decay of the signal ($< 1$ $\mu$s). Given the $^{29}$Si concentration, about $10\%$ of the donors have one or more $^{29}$Si with coupling $> 300$ kHz in the proximity, which therefore leads to a rapid decay of the total echo signal within $\sim 1$ $\mu$s by about $10\%$. In the experimental data, this fast decay is not visible because the echo signal is measured at longer times, and therefore the ESEEM signals presented in Fig.S5 in the Supplementary Information are those from $^{29}$Si with couplings $< 200$ kHz.

As for spins with a coupling strength between 20 kHz and 200 kHz, they lead to ESEEM amplitude much less than $1\%$ as shown in Figs. S7-S9 of the SI. For nuclear spins with a hyperfine coupling $< 100$ kHz, the fictitious spin model produces results with negligible errors of the modulation frequencies from the exact solution (Figs. S5 and S6 in the Supplementary Information). Furthermore, the systematic numerical studies (Figs.S9 in the Supplementary Information) show that a nearby Si nuclear spin with coupling $< 100$ kHz has little effects on the ESEEM due to other distant nuclear spins.

Considering these different contributions of Si nuclear spins of different hyperfine couplings, as discussed in the paragraph above and in more details in the Supplementary Information, we apply the fictitious spin-1/2 model to each EPR-allowed



transition of the bismuth donor manifold, considering only Si nuclear spins that have a hyperfine coupling weaker than a certain cut-off which we choose as 20 kHz, and discarding all the others.

For each transition, we compute the hyperfine parameters that enter the fictitious spin-1/2 model for all sites of the silicon lattice. We then generate a large number of random configurations of nuclear spins. We compute the corresponding 2-, 3-, or 5-pulse ESEEM signal using the analytical formulas of section IID after discarding all nuclei whose hyperfine coupling is larger than 20 kHz. We average the signal for one configuration over all bismuth donor transitions using the weights determined by simulation, and then average the results over all the configurations computed. In this way, we obtain the curves shown in Fig.9.

We use the two-pulse-Echo dataset to determine the most likely sample concentration in $^{29}$Si, using $p$ as a fitting parameter. The best fit is obtained for $p = 4.4 \pm 1 \times 10^{-4}$, which is compatible with the specified $5 \times 10^{-4}$. The agreement is satisfactory but not perfect, as seen for instance in the amplitude of the short-time ESEEM oscillations which are lower in the measurements than in the simulations, particularly at larger field. Also, the peak at $2\omega_I$ is notably broader and has a lower amplitude than in the experiment.

For the fitted value of $p$, the 3- and 5-pulse theoretical signals are also computed, and found to be in overall agreement with the data, even though the decay of the ESEEM signal predicted by the model is faster than in the experiment, and correspondingly the predicted ESEEM spectrum broader than the data.

## 6   Conclusion

We have reported 2-, 3- and 5-pulse ESEEM measurements using a quantum-limited EPR spectrometer on two model systems: erbium ions in a $CaWO_4$ matrix, and bismuth donors in silicon. Whereas the erbium measurements are done in a commonly used regime of high field, the bismuth donor measurements are performed in an unusual regime of low nuclear spin density, low hyperfine coupling, and almost zero magnetic field. Good agreement is found with the simplest analytical ESEEM models. Our results demonstrate that quantum-limited EPR spectroscopy at millikelvin temperatures can be performed with sufficient bandwidth to detect ESEEM without compromising the high spin detection sensitivity.

**Code and data availability**

All code and data necessary for generating figures 6-11 can be found at
https://doi.org/10.7910/DVN/ZJ2EEX. The analysis and plotting code is written in Python (.py) and Igor (.pxp). These files are sorted according to figure number, with the relevant files for each figure compressed into a single 7zip file (.7z).

**Author Contributions**

S.P., M.R., M.L.D, A.D., and P.B. planned and designed the experiment. Z.Z., P.G. prepared the Er:CaWO$_4$ crystal. S.P., M.R., and M.L.D. fabricated the devices, set up the experiment, and acquired the data. S.P., G.L.Z, M.R., V.R., M.L.D., B.A., A.D., R.B.L., T.C., P.G., P.B. worked on the data analysis. The project was supervised by R.B.L. and P.B. All authors contributed to manuscript preparation.

**Acknowledgements**

We thank P. Sénat, D. Duet and J.-C. Tack for the technical support, and are grateful for fruitful discussions within the Quantronics group. We acknowledge IARPA and Lincoln Labs for providing a Josephson Traveling-Wave Parametric Amplifier used in some of the measurements. We acknowledge support of the European Research Council under the European Community's Seventh Framework Programme (FP7/2007-2013) through grant agreement No. 615767 (CIRQUSS) and under the European Union's Horizon 2020 research and innovation programme [Grant Agreement No. 771493 (LOQO-MOTIONS)], of the Agence Nationale de la Recherche under the Chaire Industrielle NASNIQ (grant number ANR-17-CHIN-0001), the project QIPSE (Hong Kong RGC – French ANR Joint Scheme Fund Project A-CUHK403/15), and the project MIRESPIN, and of Region Ile-de-France Domaine d'Interet Majeur SIRTEQ under grant REIMIC. MR acknowledges a Marie Curie Individual Grant of the European Union (grant 792727, Project SMERC). AD acknowledges a SNSF mobility fellowship (177732).



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
