# Peer review of "Hyperfine spectroscopy in a quantum-limited spectrometer"

_Magnetic Resonance, 2020_

## Referee Comment (RC1) · Graham Smith (Referee) · 13 Aug 2020

General

I think this is a high quality paper that builds on the groups previous published work. The combination of tiny resonant volume, very high Q, ultra-low temperature, but still relatively short T1 (due to the Purcell effect) , and very low detector noise figure, I think previously set a new standard for absolute sensitivity for inductively detected EPR, and I think is a terrific technical achievement.

I think this paper now convincingly shows that the previous system bandwidth can be widened sufficiently to enable quantitative ESEEM to be demonstrated on two model systems (in an interesting rather low field regime) and supports the statement that

"the results bring quantum limited EPR spectroscopy one step closer to real world applications". In my view, that is a very nice stand-alone result.

I think the paper will be of interest to those involved with instrumentation, quantum control, and solid-state EPR more generally.

At one level, with minor changes the paper could be published as is, but I have a few suggestions regarding content and structure for the authors. I stress that the comments below are only intended to require minor changes and not require substantial extra work. However, I think a few extra sentences in key places would add to the impact and understanding of the work for the target audience for this journal.

Specific Comments

I might consider slightly rearranging the introduction. It perhaps was not intended, but the implied criticism of non-induction mode techniques in the introduction is that they often require specific samples and special instrumentation, but then the authors then describe a very special system that requires specially prepared samples and (at first sight) specific spectroscopic properties.

The authors state that normal ESR systems have spin sensitivities of between $10^6$ and $10^{11}$ per root Hz (and have previously quoted $10^{13}$ in their previous Nature publication). This statement really needs some references to allow fair comparisons, as sensitivity not only depends on frequency and temperature (and type of resonator), as indeed the authors point out, but also very much on sample linewidth and relaxation times (and for ESEEM the hyperfine splittings) Some of these comparisons I suspect are also for systems more optimised for concentration sensitivity. It would also be appropriate to reference other groups that have also done nice work on improving absolute sensitivity (such as Blank and Suter).

I really wasn't sure about just repeating the background ESEEM theory from Jeschke and Schweiger at the start of Section 2, as opposed to giving key formulae and referencing Schweiger. Most pulse EPR spectroscopists will be familiar with that theory, or if they are not – would almost certainly gain more form a more qualitative overview of ESEEM, and then just giving the key formulae and references. It wasn't clear to me that it was particularly helpful in understanding later discussions, at least in the main text, and so could perhaps be put in the SI. I would comment there is a lot of nice work / studies described within the SI where I think it is helpful. This is just a suggestion, and I wouldn't object if the authors felt strongly otherwise.

However, I think most EPR spectroscopists would prefer to have a slightly more extended discussion on what the barriers are to extending the technique to more "real-world samples", (as mentioned at the end of the introduction). I found it a little difficult to judge how far the instrumentation could be pushed towards the bandwidths typically used in standard experiments with commercial spectrometers or what the limitations were compared to the authors previous published results.

I think most EPR spectroscopists, when they think of EPR at very cold temperatures, think of the problems associated with very long T1's. In this specific case, I believe, remarkably, T1's are kept relatively short by the Purcell effect. I know it is clear from previous publications but I think this could be emphasized more prominently in the text. More importantly it wasn't clear to me whether the changes in the resonator to accommodate wider bandwidths had an effect on the Purcell rate and the T1 (and repetition rate).

In that context it would also be helpful to state the T1's of both systems and the T2 of the Erbium system, If they are known. At one point it is mentioned that discussion of T2 for is left to another publication, but the value could still be stated.

Similarly, ultra-low noise cryogenic amplifiers are usually avoided in pulse ESR systems, as they easily saturate or are damaged too easily or limit dynamic range. This aspect is partly covered in previous publications but is not really mentioned in the text – and here is likely to be important as input power is increased as bandwidth is lowered

or shaped pulses are being used. What is the power level of the pulses and what is a practical limit in terms of saturating amplifiers or putting too much heat into the resonator? I recall from a previous publication that the first pre-amplifier (JPA) is switched off during the pulses, but then how long does it take to switch on and what is the saturation and damage threshold. The same question also applies to the following HEMT pre-amp.

I would also find it helpful to give an effective conversion factor (Gauss per root W) for each resonator, which is more common in the EPR literature. Some papers normalise this to the Q. Others do not. As long as it is clear it does not matter.

A major instrumental concern with very cold systems is usually the heat load from the transmission line that transmits the microwaves. I'm guessing the antenna mentioned is some form of dielectric line which gives the necessary thermal insulation. A brief explanation and reference would be helpful.

Most commercial spectrometers are optimised for concentration sensitivity. Normally I think of systems that have very good absolute sensitivity as having rather poor concentration sensitivity (and vice-versa) but perhaps the difference is not so great here, as the spectrometer operates in an unusual regime. Is it possible to make a fair comparison on concentration sensitivity?

Is it the authors view that the technique is restricted to narrow-line semi-conductor and single crystals with specific spectroscopic properties. It potentially looks interesting for looking at paramagnetic defects at, or near surfaces. Is it possible/plausible to think of a scanning superconducting resonant probe rather than placing the resonator directly on the surface of interest? Is it possible to estimate the required surface density of spins for this to be viable?

I think it would be helpful if the authors could say something, even qualitative in the discussion about practical requirements on the linewidth, the hyperfine splittings, the range of concentration of spins within the effective resonant volume and the maximum

practical magnetic fields (relating to the choice of superconducting resonator).

Who did the work?

Possibly an oversight but I noticed John Morton does not appear to be assigned to any of the work packages mentioned, (other than all contributed to the writing).

Typographical.

The paper is well written.

I note, there are inconsistencies regarding spaces between numbers and units, and spaces before brackets, but that doubtless will be sorted on final type-setting.

I think, where possible, figure captions should be self contained, and it thus would be helpful to be slightly more explicit about which system is being measured within each caption.

Overall

Overall, I found the paper very interesting and recommend publication after minor changes.

---

## Referee Comment (RC2) · Edgar Groenen (Referee) · 18 Aug 2020

The manuscript by Probst et al. reports the observation of electron-spin-echo-envelope modulations on their so-called quantum-limited spectrometer. As a follow-up to previous publications on this special high-sensitivity system, the authors demonstrate that they have realized the bandwidth required to detect the echo modulation owing to the interaction of an electron spin with very weakly coupled nuclear spins. This represents a next successful step in quantum-limited EPR and as such deserves publication.

As this referee report is part of a 'discussion' and follows the report by Graham Smith, I first note that I fully agree with his remarks/suggestions/questions. Taking these into consideration will for sure enhance the impact of the paper. This particularly applies to

the proposed discussion of the technical developments that are possible/ necessary/ foreseen. Doing so, the EPR community may be able to judge how close they (we) are to what the authors refer to as 'real-world applications'.

To clarify this point, I emphasize that ESEEM is a hyperfine spectroscopy. In other words, ESEEM is applied to probe the electronic wave function through the detection of the interaction of the electron spin with nuclear spins. For the two examples treated here, the erbium-doped CaWO4 and the bismuth donor in silicon, the observed modulations provide no information on the extent of the electronic wave function. For the erbium case, the erbium spin is (taken) completely localized on the erbium center, and the resolved ESEEM frequencies are determined by the positions of the tungsten nuclei, which are known from the crystal structure. For the bismuth case, modulations are at the Larmor frequency and refer to very weakly coupled silicon nuclei, so-called 'matrix nuclei', for which the ESEEM frequency does not deviate enough from the Larmor frequency to be resolved. In other words, these two examples suffice to demonstrate that ESEEM is feasible with this spectrometer, but do not generate any new information on the electronic wave function, i.e., do not fulfil the goal of hyperfine spectroscopy. These considerations are not meant to criticize the research, but to convince the authors that a discussion of the potential of the technique, in particular of the bandwidth that can be reached, may help to increase the impact of the paper.

Some minor points. 1. The authors might consider to shorten sections 2.1 to 2.3, because the description on page 2 to page 6 is standard and can be found in many textbooks. 2. For erbium, $m_I$ is described as good quantum number, which is not obvious for the experiments at the lower magnetic fields (cf. fig 3). 3. Sections numbered Arabic are referred to in the text by Roman numbers. 4. For the modeling of the data in fig. 7, the magnitude of the magnetic field is taken as a parameter. How do the resulting values compare with the experimental values? 5. To which experiment do the $\chi2$ values in fig.9b refer? 6. Some numbers in figures may benefit from a larger font, e.g. in figures 4 and 11.

---

## Author Response (AR1)

I think this is a high quality paper that builds on the groups previous published work. The combination of tiny resonant volume, very high Q, ultra-low temperature, but still relatively short T1 (due to the Purcell effect) , and very low detector noise figure, I think previously set a new standard for absolute sensitivity for inductively detected EPR, and I think is a terrific technical achievement.

I think this paper now convincingly shows that the previous system bandwidth can be widened sufficiently to enable quantitative ESEEM to be demonstrated on two model systems (in an interesting rather low field regime) and supports the statement that C1 MRD Interactive comment Printer-friendly version Discussion paper "the results bring quantum limited EPR spectroscopy one step closer to real world applications".

In my view, that is a very nice stand-alone result. I think the paper will be of interest to those involved with instrumentation, quantum control, and solid-state EPR more generally.

*We gratefully thank Graham Smith for his kind words and are especially happy that experienced EPR spectroscopists find our work of interest.*

At one level, with minor changes the paper could be published as is, but I have a few suggestions regarding content and structure for the authors. I stress that the comments below are only intended to require minor changes and not require substantial extra work. However, I think a few extra sentences in key places would add to the impact and understanding of the work for the target audience for this journal.

Specific Comments

I might consider slightly rearranging the introduction. It perhaps was not intended, but the implied criticism of non-induction mode techniques in the introduction is that they often require specific samples and special instrumentation, but then the authors then describe a very special system that requires specially prepared samples and (at first sight) specific spectroscopic properties.

*We agree with M. Smith's comment. To avoid any implicit criticism, we remove the few words on specific samples and special instrumentation in the revised manuscript. See page 1 (lines 25, 26 & 28)*

The authors state that normal ESR systems have spin sensitivities of between $10^6$ and $10^{11}$ per root Hz (and have previously quoted $10^{13}$ in their previous Nature publication). This statement really needs some references to allow fair comparisons, as sensitivity not only depends on frequency and temperature (and type of resonator), as indeed the authors point out, but also very much on sample linewidth and relaxation times (and for ESEEM the hyperfine splittings) Some of these comparisons I suspect are also for systems more optimised for concentration sensitivity. It would also be appropriate to reference other groups that have also done nice work on improving absolute sensitivity (such as Blank and Suter).

*We agree with M. Smith that we should be more precise in our statements about spin detection sensitivity, and more complete in our references. We have attempted to do so in the revised manuscript, with more references being been added on page 1 (lines 27, 31, 37 & 37) and also page 2 (line 3)*

I really wasn't sure about just repeating the background ESEEM theory from Jeschke and Schweiger at the start of Section 2, as opposed to giving key formulae and referencing Schweiger. Most pulse EPR spectroscopists will be familiar with that theory, or if they are not – would almost certainly gain more form a more qualitative overview of ESEEM, and then just giving the key formulae and references. It wasn't clear to me that it was particularly helpful in understanding later discussions, at least in the main text, and so could perhaps be put in the SI.

I would comment there is a lot of nice work / studies described within the SI where I think it is helpful. This is just a suggestion, and I wouldn't object if the authors felt strongly otherwise.

*We understand the point raised by M. Smith. Because several of the equations are used in later parts of the paper, we decided not to remove the entire section 2 from the main text. However, we shortened Section 2.2, by removing all equations that were not strictly necessary for the ESEEM formulas (see page 4 and page 5). We hope this shorter version will be more satisfactory.*

However, I think most EPR spectroscopists would prefer to have a slightly more extended discussion on what the barriers are to extending the technique to more "realworld samples", (as mentioned at the end of the introduction). I found it a little difficult to judge how far the instrumentation could be pushed towards the bandwidths typically used in standard experiments with commercial spectrometers or what the limitations were compared to the authors previous published results. I think most EPR spectroscopists, when they think of EPR at very cold temperatures, think of the problems associated with very long T1's. In this specific case, I believe, remarkably, T1's are kept relatively short by the Purcell effect. I know it is clear from previous publications but I think this could be emphasized more prominently in the text. More importantly it wasn't clear to me whether the changes in the resonator to accommodate wider bandwidths had an effect on the Purcell rate and the T1 (and repetition rate).

*As rightfully expressed by Graham Smith, the Purcell rate depends on the resonator bandwidth $\kappa$ as $\Gamma_P = 4g^2/\kappa$, g being the spin-photon coupling constant. For systems that are in the Purcell limit, $T_1 = \Gamma_P^{-1}$, and increasing $\kappa$ (for the purpose of ESEEM spectroscopy or other) has indeed a « double effect » which is unusual in EPR spectroscopy : it leads to reduced sensitivity (usual) but also longer T1 (less usual). This concept has now been introduced on page 2 (lines 8 - 11) and is explained in the discussion/conclusion section of the revised manuscript on page 18 (lines 12-18)*

In that context it would also be helpful to state the T1's of both systems and the T2 of the Erbium system, If they are known. At one point it is mentioned that discussion of T2 for is left to another publication, but the value could still be stated.

*Agreed. T1 has been mentioned on page 12 (line 10) and T2 has been mentioned on page 13 (line 19)*

Similarly, ultra-low noise cryogenic amplifiers are usually avoided in pulse ESR systems, as they easily saturate or are damaged too easily or limit dynamic range. This aspect is partly covered in previous publications but is not really mentioned in the text – and here is likely to be important as input power is increased as bandwidth is lowered or shaped pulses are being used. What is the power level of the pulses and what is a practical limit in terms of saturating amplifiers or putting too much heat into the resonator? I recall from a previous publication that the first pre-amplifier (JPA) is switched off during the pulses, but then how long does it take to switch on and what is the saturation and damage threshold. The same question also applies to the following HEMT pre-amp.

*Saturation is a concern at the level of the superconducting amplifiers, in particular whenever control pulses are applied. Fortunately, the amplifiers recover quickly (within few microseconds), after what they can amplify the spin-echoes without problem. We occasionally switch off some amplifiers (the JPAs) by switching off the pump tone, and others not (JTWPA); we find that this is not a major requirement. The following HEMT amp is never saturated, so there is no problem there.*

*We have added these aspects in the revised manuscript on page 10 (lines 5-11)*

I would also find it helpful to give an effective conversion factor (Gauss per root W) for each resonator, which is more common in the EPR literature. Some papers normalise this to the Q. Others do not. As long as it is clear it does not matter. A major instrumental concern with very cold systems is usually the heat load from the transmission line that transmits the microwaves. I'm guessing the antenna mentioned is some form of dielectric line which gives the necessary thermal insulation. A brief explanation and reference would be helpful.

*We provide the conversion factor, with its definition, in the revised manuscript on pages 10 (line 20) for the Er:CaWO4 system and on page 11 (line 13) for the Bi:Si system.*

*In our system, the microwaves are delivered via coaxial cables. We use superconducting cables between 4K and 10mK made out of Niobium Titanium. They provide lossless microwave transmission, but they do not conduct heat. These are commonly employed in circuit QED setups. We included this point on page 9 (lines 28-31) of the revised manuscript.*

Most commercial spectrometers are optimised for concentration sensitivity. Normally I think of systems that have very good absolute sensitivity as having rather poor concentration sensitivity (and vice-versa) but perhaps the difference is not so great here, as the spectrometer operates in an unusual regime. Is it possible to make a fair comparison on concentration sensitivity?

*We are not quite sure how to do this comparison. We respectfully point out that we provide the absolute sensitivity as well as the detection mode volume, so people interested in concentration*

*sensitivity at least have all the elements to do a quick estimate. We would prefer not to enter into this consideration if the referee agrees, as we believe it is significantly outside the scope of our work.*

Is it the authors view that the technique is restricted to narrow-line semi-conductor and single crystals with specific spectroscopic properties. It potentially looks interesting for looking at paramagnetic defects at, or near surfaces.

*Although we have so far used the technique only with narrow-line semiconductor defects, we don't think it is in principle restricted to these specific systems. Indeed, surface defects more generally are clearly an interesting area of applications. However, we have not included any discussion of this subject in the revised manuscript, as we feel that it is beyond the scope of this work.*

Is it possible/plausible to think of a scanning superconducting resonant probe rather than placing the resonator directly on the surface of interest?

*It is indeed likely to be possible, and would certainly be a very interesting project. Once again, however, we have not included any discussion of this subject in the revised manuscript as we feel that it is not directly relevant within the scope of this work.*

Is it possible to estimate the required surface density of spins for this to be viable? I think it would be helpful if the authors could say something, even qualitative in the discussion about practical requirements on the linewidth, the hyperfine splittings, the range of concentration of spins within the effective resonant volume and the maximum practical magnetic fields (relating to the choice of superconducting resonator).

*We have included a prospective discussion regarding this point on page 18 (lines 16-20) of the revised manuscript*

Who did the work?

Possibly an oversight but I noticed John Morton does not appear to be assigned to any of the work packages mentioned, (other than all contributed to the writing).

*John Morton also prepared and provided the bismuth-donor-implanted silicon sample. We simply forgot to mention this contribution. This is corrected in the revised manuscript on page 18 (line 27)*

Typographical.

The paper is well written. I note, there are inconsistencies regarding spaces between numbers and units, and spaces before brackets, but that doubtless will be sorted on final type-setting. I think, where possible, figure captions should be self contained, and it thus would be helpful to be slightly more explicit about which system is being measured within each caption.

*We agree with Graham Smith and have accordingly modified the caption of Figs.9,10,11 which indeed did not refer explicitly to the system that was being measured (Bi :Si donors)*

Overall Overall, I found the paper very interesting and recommend publication after minor changes.

The manuscript by Probst et al. reports the observation of electron-spin-echo-envelope modulations on their so-called quantum-limited spectrometer. As a follow-up to previous publications on this special high-sensitivity system, the authors demonstrate that they have realized the bandwidth required to detect the echo modulation owing to the interaction of an electron spin with very weakly coupled nuclear spins. This represents a next successful step in quantum-limited EPR and as such deserves publication.

*We thank Edgar Groenen for his positive assessment of our work.*

As this referee report is part of a 'discussion' and follows the report by Graham Smith, I first note that I fully agree with his remarks/suggestions/questions. Taking these into consideration will for sure enhance the impact of the paper. This particularly applies to the proposed discussion of the technical developments that are possible/ necessary/ foreseen. Doing so, the EPR community may be able to judge how close they (we) are to what the authors refer to as 'real-world applications'.

*We have indeed attempted to make our article more complete. Additional discussion regarding the 'closeness' of this work to real-world applications has been included on page 18 (lines 7-20)*

To clarify this point, I emphasize that ESEEM is a hyperfine spectroscopy. In other words, ESEEM is applied to probe the electronic wave function through the detection of the interaction of the electron spin with nuclear spins. For the two examples treated here, the erbium-doped CaWO4 and the bismuth donor in silicon, the observed modulations provide no information on the extent of the electronic wave function. For the erbium case, the erbium spin is (taken) completely localized on the erbium center, and the resolved ESEEM frequencies are determined by the positions of the tungsten nuclei, which are known from the crystal structure. For the bismuth case, modulations are at the Larmor frequency and refer to very weakly coupled silicon nuclei, so-called 'matrix nuclei', for which the ESEEM frequency does not deviate enough from the Larmor frequency to be resolved. In other words, these two examples suffice to demonstrate that ESEEM is feasible with this spectrometer, but do not generate any new information on the electronic wave function, i.e., do not fulfil the goal of hyperfine spectroscopy. These considerations are not meant to criticize the research, but to convince the authors that a discussion of the potential of the technique, in particular of the bandwidth that can be reached, may help to increase the impact of the paper.

*We thank the reviewer for clarifying this point, and we believe to have followed these suggestions in the revised manuscript. In particular, we have added a discussion regarding potential bandwidths of this technique on page 18 (lines 13-20)*

Some minor points. 1. The authors might consider to shorten sections 2.1 to 2.3, because the description on page 2 to page 6 is standard and can be found in many textbooks.

*We have indeed shortened section 2.2 in the revised manuscript; see changes on pages 4 and 5.*

2. For erbium, mI is described as good quantum number, which is not obvious for the experiments at the lower magnetic fields (cf. fig 3).

*We agree with the reviewer that m_I is only an approximate quantum number, because we are not deep into the high field limit. We have modified the text accordingly on page 7 (lines 38-40) and page 8 (lines 1-2).*

3. Sections numbered Arabic are referred to in the text by Roman numbers.

*Corrected in the revised manuscript on pages 6-9 & 12-17*

4. For the modeling of the data in fig. 7, the magnitude of the magnetic field is taken as a parameter. How do the resulting values compare with the experimental values?

*Since our coils are home-made, the calibration also has some degree of uncertainty. The resulting values from the fit agree with the calibrated value, within this uncertainty and this has been stated on page 13 (line 20) of the revised manuscript.*

5. To which experiment do the $\chi 2$ values in fig.9b refer?

*The chi2 in fig.9b refers to the fit of the relative concentration of Si29 nuclei, using the 2-pulse ESEEM data. This is explained in the section "Comparison with the model" (5.2.4). In the revised manuscript, we have added an explicit reference to figure 9b, showing the chi2 of the fit.*

6. Some numbers in figures may benefit from a larger font, e.g. in figures 4 and 11.

*Figures 4 and 11 were scaled incorrectly (they were too small) because they are two-column figures. They have been rescaled in the revised manuscript.*

[revised manuscript text omitted]

where $H_{\text{e}} = \omega_S S_z$ ($H_{\text{n}} = \omega_I I_z$) is the Zeeman Hamiltonian of the electron (nuclear) spin with Larmor frequency $\omega_S$ ($\omega_I$), and $H_{\text{hf}}$ is the electron-nuclear hyperfine interaction, which includes their dipole-dipole coupling and may include a Fermi contact term as well. We assume that $\omega_S$ is much larger than the hyperfine interaction strength, in which case terms proportional to the $S_x$ and $S_y$ operators can be neglected. This secular approximation leads to a hyperfine Hamiltonian of the form $H_{\text{hf}} = AS_zI_z + BS_zI_x$, with the expressions for $A$ and $B$ depending on the details of the hyperfine interaction(Schweiger and Jeschke, 2001).

Overall, the system Hamiltonian is

$$H_0 = \omega_S S_z + \omega_I I_z + AS_z I_z + BS_z I_x. \tag{2}$$

Because of the $BS_zI_x$ term, the nuclear spin is subjected to an effective magnetic field whose direction (and magnitude) depend on the electron spin state $|\uparrow_{\text{e}}\rangle$ or $|\downarrow_{\text{e}}\rangle$. Its eigenstates therefore depend on the electron spin state, so that nuclear-spin-non-preserving transitions become allowed , which leads between all the spin system energy levels $|1\rangle - |4\rangle$, leading to the ESEEM phenomenon.

More precisely, the Hamiltonian Eq.2 can be diagonalized leading to the following four eigenstates

[Figure]

**Figure 2.** ESEEM model system for electron spin $S = 1/2$ and nuclear spin $I = 1/2$ with $\omega_I, A, B > 0$. (a) Nuclear spin (purple) subject to external field $B_0$ and dipole field (blue) of a nearby electron spin (green) located at relative position $\boldsymbol{r}$. (b) Energy diagram showing the electron transitions (green), the nuclear transitions (purple), and the (normally forbidden) electro-nuclear transitions (orange). The energy levels $|1\rangle, ..., |4\rangle$ are labeled according to the eigenstates of the Zeeman basis. (c) Quantization axes $\omega_\uparrow$ and $\omega_\downarrow$ due to mixing of the nuclear states, which results in inclination of the quantization axis from $z$ by the angles $\eta_\uparrow$ and $\eta_\downarrow$, respectively. (d) EPR spectrum showing the electron transitions (green) and the electro-nuclear transitions (orange) as well as the relation of these ESR transitions to the nuclear frequencies $\omega_\uparrow$ and $\omega_\downarrow$ (purple).

$$|1\rangle = | \uparrow_e\rangle(\cos\frac{\eta_\uparrow}{2} \uparrow_n\rangle + \sin\frac{\eta_\uparrow}{2}|\downarrow_n\rangle)$$

$$|2\rangle = | \uparrow_e\rangle(\sin\frac{\eta_\uparrow}{2}|\uparrow_n\rangle - \cos\frac{\eta_\uparrow}{2}|\downarrow_n\rangle)$$

$$|3\rangle = | \downarrow_e\rangle(\cos\frac{\eta_\downarrow}{2}|\uparrow_n\rangle + \sin\frac{\eta_\downarrow}{2}|\downarrow_n\rangle)$$

$$|4\rangle = | \downarrow_e\rangle(\sin\frac{\eta_\downarrow}{2}|\uparrow_n\rangle - \cos\frac{\eta_\downarrow}{2}|\downarrow_n\rangle),$$

where subscript e (resp. n) refers to the electron (resp. nuclear) state, and

$$\eta_\uparrow = \arctan\frac{B}{A + 2\omega_I}$$

$$\eta_\downarrow = \arctan\frac{B}{A - 2\omega_I}.$$

Physically, $\eta_{\uparrow,\downarrow}$ is the Relevant parameters are the electron-spin-state-dependent angle angles between the effective magnetic field seen by the nuclear spin and the quantization axis $z$ . The energies of these states are

$$\epsilon_1 \eta_\uparrow = \frac{\omega_S}{2} + \frac{\omega_\uparrow}{2}\arctan\frac{B}{A + 2\omega_I}$$

$$\epsilon_2 \eta_\downarrow = \frac{\omega_S}{2} - \frac{\omega_\uparrow}{2} \quad \epsilon_3 = -\frac{\omega_S}{2} + \frac{\omega_\downarrow}{2} \quad \epsilon_4 = -\frac{\omega_S}{2} - \frac{\omega_\
[revised manuscript text omitted]

---

## Author Response (AR2)

**Authors response (Marked up PDF starts on Page 3) :**

The Authors would like to thank the Editor's decision to accept our Manuscript for publication, and would also like to thank Graham Smith for his detailed review of our work and useful suggestions.

*Main text: In the version I was sent there were some "?" I suspect left over from the editing process in the paragraph after Fig 11.*

These were due to a missing reference (the replication data repository) and this has now been corrected.

*In Fig 10 there seem to be a problem with the scale by the side of the graph.*

The missing scale has now been fixed.

*In Fig 11 the text in the graph is very much smaller than other graphs.*

This figure will be two-column in the published version, at which point the text will be scaled as the other figures.

*There is a Gap after section 5.2.4.*

The gap has been removed

*There is still some some inconsistencies as to whether a gap is left between numbers and units - and the whole document should be checked for those.*

All numbers are now followed by a gap for consistency, and units have been modified to the in-house style.

*The very last sentence in the discussion does not scan well.*

The last two sentences have been re-arranged and re-worded for clarity.

*SI:I don't think formatting is so critical in the SI, but as they have put so much work into it, they might choose to have the Table caption consistently either above or below the Table. Regardless I would have a line separating the table and the caption for Table S3 and S4.*

All captions are now above their respective tables, and a line has been included to separate the caption and table for S3 & S4

*I didn't understand why the references were on P.12 with most, but not all, of the figures following - unless it is a house style thing. I would prefer to see them integrated, but if it is common to do otherwise....*

The figures have now been integrated

*Its possible I have missed it but also didn't see Figure S7 mentioned in the main text.*

Fig. S7 has been mentioned in the text (page 13, line 1 of the current version)

[revised manuscript text omitted]